# Shared mechanisms of auditory and non-auditory vocal learning in the songbird brain

**James N McGregor**[1]*[†], **Abigail L Grassler**[2][†], **Paul I Jaffe**[3], **Amanda Louise Jacob**[2], **Michael S Brainard**[3,4], **Samuel J Sober**[2]

[1]Neuroscience Graduate Program, Graduate Division of Biological and Biomedical Sciences, Laney Graduate School, Emory University, Atlanta, United States; [2]Department of Biology, Emory University, Atlanta, United States; [3]Center for Integrative Neuroscience, University of California, San Francisco, San Francisco, United States; [4]Howard Hughes Medical Institute, University of California, San Francisco, San Francisco, United States

**Abstract** Songbirds and humans share the ability to adaptively modify their vocalizations based on sensory feedback. Prior studies have focused primarily on the role that auditory feedback plays in shaping vocal output throughout life. In contrast, it is unclear how non-auditory information drives vocal plasticity. Here, we first used a reinforcement learning paradigm to establish that somatosensory feedback (cutaneous electrical stimulation) can drive vocal learning in adult songbirds. We then assessed the role of a songbird basal ganglia thalamocortical pathway critical to auditory vocal learning in this novel form of vocal plasticity. We found that both this circuit and its dopaminergic inputs are necessary for non-auditory vocal learning, demonstrating that this pathway is critical for guiding adaptive vocal changes based on both auditory and somatosensory signals. The ability of this circuit to use both auditory and somatosensory information to guide vocal learning may reflect a general principle for the neural systems that support vocal plasticity across species.

*For correspondence: jmcgregor2292@gmail.com

[†]These authors contributed equally to this work

Competing interest: The authors declare that no competing interests exist.

## Editor's evaluation

This is an important article that shows that songbirds can learn to adjust their song on the basis of somatosensory feedback, and not just auditory feedback as previously thought. Convincing evidence is provided that cutaneous stimulation-induced song learning requires the same dopamine-basal ganglia pathway previously implicated in natural auditory feedback-based learning, showing that vocal production circuits can flexibly learn from feedback from multiple modalities.

## Introduction

A fundamental goal of neuroscience is to understand how the brain uses sensory feedback to drive adaptive changes in motor output (*Graybiel et al., 1994*; *Hikosaka et al., 2002*). Human speech is a prime example of a sensory-guided behavior, and humans are among the few species that use auditory feedback from their own vocalizations to compensate for perceived errors in vocal output (*Doupe and Kuhl, 1999*). This reliance on sensory feedback for speech production is lifelong: loss of hearing impairs both speech development and vocal production in adulthood, and adult speakers rely heavily on auditory signals to calibrate their vocal acoustics (*Oller and Eilers, 1988*; *Cowie and Douglas-Cowie, 1983*; *Stoel-Gammon and Otomo, 1986*; *Houde and Jordan, 1998*). Accordingly, studies of the neurobiology of speech have focused on the specialized neural pathways that process

**Figure 1.** Schematic of songbird neural circuitry and hypotheses tested. (**a**) Sagittal schematic view of songbird brain circuitry. Brain nuclei of the motor pathway – the neural circuit for vocal production – are black. Brain nuclei of the anterior forebrain pathway (AFP) – the neural circuit for vocal learning – are red. Ventral Tegmental Area (VTA, shown in purple) provides dopaminergic input into Area X, the basal ganglia nucleus of the AFP. (**b**) The three primary hypotheses tested in this article. In the first set of experiments, we tested whether non-auditory input can drive adaptive changes to adult song (Experiment 1). In the second set of experiments, we assessed the necessity of LMAN for non-auditory vocal learning (Experiment 2). In the third set of experiments, we tested the necessity of dopaminergic projections to Area X for non-auditory vocal learning (Experiment 3).

auditory feeback (*Jarvis, 2019*). In contrast, it is unclear whether the brain uses non-auditory sensory input to modulate the acoustics of vocal production, although studies demonstrating that humans use non-auditory (somatosensory) signals to calibrate jaw movements suggest that this might be the case (*Nasir and Ostry, 2008*; *Tremblay et al., 2003*).

We address how the brain processes different sources of sensory feedback to guide vocal behavior by using a model system ideally suited for the study of vocal learning, the Bengalese finch. Like humans, songbirds rely on auditory signals to precisely calibrate their vocal output throughout life (*Sober and Brainard, 2009*; *Kuebrich and Sober, 2015*; *Konishi, 1965*; *Nordeen and Nordeen, 1992*). Also similar to humans, songbirds have evolved specialized neural pathways for vocal learning, allowing the precise interrogation of the brain mechanisms of song plasticity (*Jarvis, 2019*; *Brainard and Doupe, 2002*). However, prior research on this brain network has focused almost exclusively on the role of auditory feedback, although recent work has shown the importance of visual cues (light) in shaping vocalizations (*Veit et al., 2021*; *Zai et al., 2020*). Previous studies have revealed that songbird brains have a basal ganglia-thalamocortical circuit, the anterior forebrain pathway (AFP), that is required for auditory-guided vocal learning but not vocal production (*Figure 1a*; *Brainard and Doupe, 2000*; *Nordeen and Nordeen, 1993*; *Mooney, 2009*; *Bottjer et al., 1984*). For example, lesions of LMAN (the output nucleus of the AFP) prevent adult vocal plasticity in response to perturbations of auditory feedback (*Brainard and Doupe, 2000*; *Ali et al., 2013*; *Morrison and Nottebohm, 1993*). Also, lesions or manipulations of dopaminergic input into Area X (the basal ganglia nucleus of the AFP) impair adult vocal learning in response to the pitch-contingent delivery of aversive auditory stimuli (white noise bursts) (*Hoffmann et al., 2016*; *Hisey et al., 2018*; *Xiao et al., 2018*). Recent work has demonstrated that the songbird AFP receives anatomical projections from brain regions that process non-auditory sensory information (*Paterson and Bottjer, 2017*), and that Area X plays a crucial role in processing visual information to shape vocal output (*Zai et al., 2020*), yet it remains

unclear whether and how the AFP processes somatosensory feedback to drive vocal learning, and whether dopaminergic input to the AFP is involved in non-auditory forms of learning.

We performed a series of three experiments (*Figure 1b*) to investigate whether and how the brain uses non-auditory, somatosensory feedback to guide vocal learning. We first tested whether adult songbirds can adaptively modify specific elements of their song structure in response to somatosensory feedback (*Figure 1b*, Experiment 1). We used non-auditory stimuli (mild cutaneous electrical stimulation), which we delivered during ongoing song performance, to differentially reinforce the acoustics (fundamental frequency, or 'pitch') of specific song elements, or 'syllables.' In separate experiments, we tested birds using auditory stimuli consisting of brief playbacks of white noise, a well-established paradigm for driving changes in pitch in adult songbirds (*Hoffmann et al., 2016*; *Andalman and Fee, 2009*; *Tumer and Brainard, 2007*). Delivering non-auditory and auditory stimuli on the same schedule therefore allowed us to directly compare how different sensory modalities affect vocal behavior. We next assessed the neural circuit mechanisms underlying somatosensory-driven vocal learning by determining the necessity of LMAN (the output nucleus of the AFP) for somatosensory learning (*Figure 1b*, Experiment 2). Finally, we assessed the role of dopaminergic neural circuitry in somatosensory vocal learning by performing selective lesions of dopaminergic input to Area X (*Figure 1b*, Experiment 3).

## Results

### Non-auditory feedback can drive adult songbird vocal learning

We tested whether non-auditory feedback can drive vocal learning (*Figure 1b*, Experiment 1) by providing mild, pitch-contingent cutaneous stimulation through a set of wire electrodes on the scalps of adult songbirds. Before initiating cutaneous stimulation training, we continuously recorded song without providing any feedback for 3 days (baseline) (*Figure 2a*). Every day, songbirds naturally produce many renditions of song, which consist of repeated patterns of unique vocal gestures, called syllables (*Figure 2b*, top). For one 'target' syllable in each experimental subject, we quantified rendition-to-rendition variability in the fundamental frequency (which we refer to here as 'pitch') of each occurrence of this syllable on the final baseline day (*Figure 2b*, top). To differentially reinforce the pitch of a target syllable, we determined a range of pitches within this baseline distribution (either all pitches above the 20th percentile or all pitches below the 80th percentile), and then triggered the delivery of cutaneous stimulation in real time (within 40 ms of syllable onset) when the pitch of the target syllable fell within this range (*Figure 2b*, bottom). We performed this pitch-contingent cutaneous stimulation training continuously for 3 days. Note that the birds could choose not to sing in order to avoid triggering any cutaneous stimulation, and we carefully monitored animal subjects for any signs of distress (see 'Materials and methods').

For example, in one experiment (shown in *Figure 2a–d*, *Figure 2—source data 1*), cutaneous stimulation was triggered on every rendition of the target syllable that had a pitch above 2.13 kHz (the 20th percentile of the baseline distribution) for 3 days. In this example experiment, the bird gradually changed the pitch of the targeted syllable downward (the adaptive direction), such that cutaneous stimulation was triggered less frequently (*Figure 2c*). In other experiments where the adaptive direction of pitch change is upward, we triggered cutaneous stimulation whenever the target syllable pitch was below the 80th percentile of this distribution. In the example experiment, at the start of the first day of cutaneous stimulation training, 80% of syllable renditions resulted in cutaneous stimulation and 20% of syllable renditions resulted in escapes. On the third (final) day of cutaneous stimulation training, escapes occurred on over 60% of target syllable renditions and the entire distribution of pitches had changed significantly in the adaptive direction, indicating that a significant amount of vocal learning occurred in this example experiment (*Figure 2d*; two-sample Kolmogorov–Smirnov test to assess the difference between baseline and end of cutaneous stimulation training, p=1.1776e-12). We then stopped triggering cutaneous stimulation and continued to record unperturbed song for six additional days (washout). After 6 days of washout, there was no significant difference between the distribution of target syllable pitches at the end of washout compared to baseline (*Figure 2d*; two-sample Kolmogorov–Smirnov test, p=0.606). For analysis of washout across all experiments, see *Figure 2—figure supplement 1*.

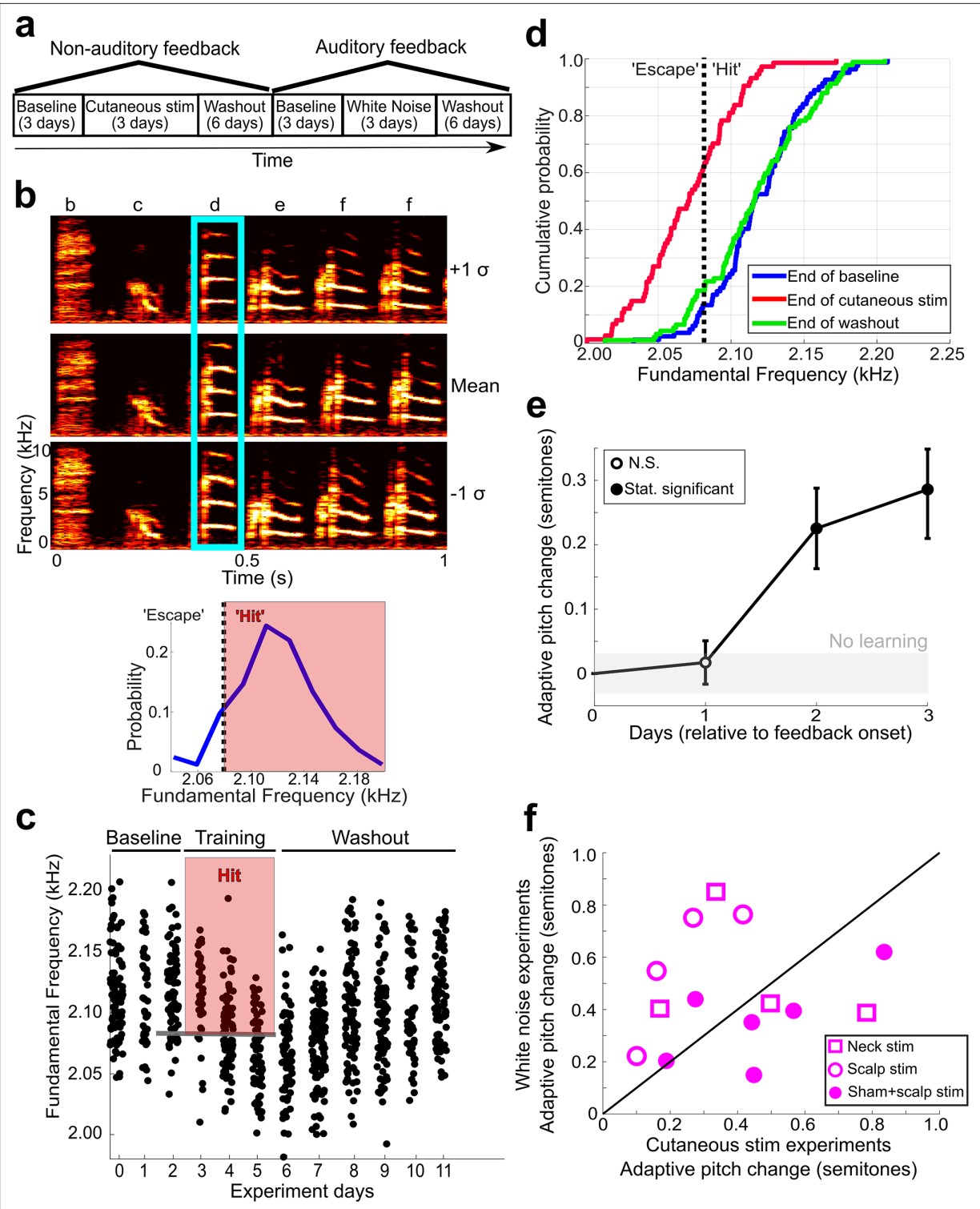

**Figure 2.** Non-auditory feedback drives vocal learning. (**a**) Timeline of vocal learning experiments in this example bird. The order of the auditory vs. non-auditory experiments was randomized across birds. (**b**) Top: spectrograms and song syllables (labeled b–f) including target syllable ('d'). Bottom: baseline pitch distribution and pitch threshold. Cutaneous stimulation was provided during renditions of the target syllable above a chosen pitch threshold ('hit'). (**c**) Each dot represents the pitch of one rendition of the target syllable. Renditions in the 'hit' range rapidly triggered a cutaneous stimulation (within 40 ms of syllable onset). During washout, cutaneous stimulation was discontinued. (**d**) Cumulative distribution function (CDF) plot showing the probability a value of pitch from a distribution falls at or below the value on the x-axis. The pitch distribution at the end of cutaneous stimulation training was significantly greater than baseline (two-sample Kolmogorov–Smirnov test, p=1.178e-12). End of washout distribution was not

*Figure 2 continued on next page*

*Figure 2 continued*

significantly different from baseline (two-sample Kolmogorov–Smirnov test, p=0.606). Panels (**b–d**) show data from the same experiment. (**e**) Adaptive pitch change (in semitones) of the target syllables during cutaneous stimulation training, grouped across 13 experiments. The mean change during training was significantly greater than baseline (the probability of resampled mean pitch on all three training days 2 and 3 lesser than or equal to zero was $P_{boot} < 0.0010$, indicated by filled circles). (**f**) Learning magnitudes (adaptive pitch change by end of training) in individual birds that underwent both white noise and cutaneous stimulation training (n = 14). Open squares indicate birds that did not undergo craniotomies for sham LMAN lesions and received cutaneous stimulation on their neck, open circles indicate birds that did not undergo craniotomies for sham LMAN lesions and received cutaneous stimulation on their scalp, and closed circles indicate birds that underwent LMAN sham operations and received cutaneous stimulation on their scalp. No significant difference in learning magnitudes during cutaneous stimulation training vs. during white noise training (paired *t*-test, p=0.313).

The online version of this article includes the following source data, source code, and figure supplement(s) for figure 2:

**Source data 1.** Source data for analyses in *Figure 2*.

**Source data 2.** Source data for analyses in *Figure 2—figure supplement 5* and *Figure 2—figure supplement 6*.

**Source data 3.** Source data for analyses in *Figure 2—figure supplement 7*.

**Source code 1.** Source code for use with *Figure 2—source data 1* for analyses in *Figure 2B-D*.

**Source code 2.** Source code for use with *Figure 2—source data 1* for analyses in *Figure 2E*.

**Source code 3.** Source code for use with *Figure 2—source data 2* for analyses in *Figure 2—figure supplement 5* and *Figure 2—figure supplement 6*.

**Source code 4.** Source code for use with *Figure 2—source data 3* for analyses in *Figure 2—figure supplement 7*.

**Figure supplement 1.** Rates of washout across different experimental conditions.

**Figure supplement 2.** Amount of pitch change on each day of cutaneous stimulation training for each individual experiment.

**Figure supplement 3.** LMAN lesions and 6-OHDA injections in Area X impair auditory-driven vocal learning.

**Figure supplement 4.** Analysis of acute effects of cutaneous stimulation on target syllable pitch.

**Figure supplement 5.** Results from 12 example non-auditory vocal learning experiments.

**Figure supplement 6.** CDF plots showing the probability a value of pitch from a distribution falls at or below the value on the x-axis for 12 example experiments from each of the birds that did not undergo brain operations.

**Figure supplement 7.** Comparison of non-auditory vocal learning when measured in the morning and in the evening.

In order to assess whether non-auditory feedback is sufficient to drive vocal learning across multiple songbirds, we first measured the adaptive pitch change (in semitones) for each individual experiment. Semitones provide a normalized measure of pitch change such that a one semitone change corresponds to a roughly 6% change in the absolute frequency of an acoustic signal (see *Equation 1*). We employed a hierarchical bootstrap approach to measure SEM and assess significance (see 'Materials and methods'; *Saravanan et al., 2020*; *Saravanan et al., 2019*) since this method more accurately quantifies the error in hierarchical data (e.g., many renditions of a target syllable collected across multiple birds). We found that the mean pitch (in semitones) of the target syllables showed a significant, adaptive change from baseline on days 2 and 3 of cutaneous stimulation training (*Figure 2e*; the probability of resampled mean pitch on cutaneous stimulation training days 2 and 3 lesser than or equal to zero was $P_{boot} < 0.0010$, limit due to resampling $10^4$ times, n = 13 experiments in 12 birds, one bird underwent two cutaneous stimulation experiments; *Figure 2—source data 1*). This demonstrates that non-auditory feedback is sufficient to drive vocal learning in adult songbirds. In all individual experiments where an upward pitch change resulted in less frequent triggering of cutaneous stimulation, the birds changed their pitch in the adaptive (upward) direction, and in all experiments where a downwards pitch change resulted in less frequent triggering of cutaneous stimulation, the birds changed their pitch in the adaptive (downward) direction (*Figure 2—figure supplement 2a*, *Figure 2—source data 1*).

To further characterize cutaneous stimulation training and compare this form of learning to well-established vocal learning paradigms, we performed multiple learning experiments – one cutaneous stimulation and one white noise – in 8 out of the 12 individual birds from this dataset where the implanted electrode wires remained intact for a long enough time to perform multiple sets of experiments (*Figure 2a*). To account for the potential influence of multiple trainings in the same individual birds on magnitude of learning, we randomized the order of white noise training and cutaneous stimulation training for the birds that underwent both training paradigms. We also included six LMAN sham-operated birds from a later set of experiments in this particular analysis. We did so because the

sham-operated birds had intact song systems and underwent both cutaneous stimulation and white noise training. Also, we found no statistically significant difference between the magnitude of learning by the end of training in birds that did not undergo craniotomies for LMAN, 6-OHDA, or sham lesions compared with the magnitude of learning in birds that received sham LMAN lesions for either white noise experiments (two-sample *t*-test, p=0.779) or cutaneous stimulation experiments (two-sample *t*-test, p=0.148).

Consistent with prior studies (*Ali et al., 2013*; *Hoffmann et al., 2016*; *Tumer and Brainard, 2007*), by the end of white noise training, the adaptive pitch change (in semitones) across all white noise experiments performed in unoperated birds (birds that had wire electrodes surgically implanted but received no invasive brain procedures like sham operations) was significantly greater than baseline (zero) (*Figure 2—figure supplement 3a*; the probability of resampled mean pitch on all three cutaneous stimulation training days lesser than or equal to zero was $P_{boot} < 0.0010$; *Figure 2—source data 1*). In the separate experimental group of birds that underwent sham operations, we also observed significant adaptive pitch changes in response to white noise bursts, as expected (*Figure 2—figure supplement 3b*; the probability of resampled mean pitch on all three cutaneous stimulation training days lesser than or equal to zero was $P_{boot} < 0.0010$; *Figure 2—source data 1*). There was significant individual variability in learning magnitudes (adaptive pitch change at the end of training) during cutaneous stimulation and white noise experiments (*Figure 2f*, *Figure 2—source data 1*). We found no systematic differences between learning magnitude during cutaneous stimulation training and the learning magnitude during white noise training (*Figure 2f*; paired *t*-test, p=0.313). These results suggest that non-auditory stimuli can drive vocal learning as effectively as auditory stimuli.

To confirm that cutaneous stimulation learning was truly driven by the non-auditory stimulus and not by an unintentional, acute change in vocal output caused by the cutaneous stimulation, we measured the syllable features of interleaved 'catch' trials, where cutaneous stimulation was randomly withheld (see 'Materials and methods') on each day of cutaneous stimulation training. For each experiment, we normalized the pitch of each catch trial from each day of training to the mean pitch of all trials where cutaneous stimulation was provided. We excluded any experiments where the total number of catch trials was less than 10. In every case, the normalized catch trials did not differ significantly from 1, indicating that the pitch of catch trials was highly similar to trials where cutaneous stimulation was provided (*Figure 2—figure supplement 4a*; *t*-test, 0.071 < p < 0.997 for each experiment; *Figure 2—source data 1*). For comparison, we also performed the same analysis on randomly selected trials from a day of baseline recording, where cutaneous stimulation was not provided on any trials (*Figure 2—figure supplement 4a*, *Figure 2—source data 1*). There was no significant difference between this dataset and the normalized catch trials (paired *t*-test, p=0.339). We repeated this analysis for other syllable features, such as syllable duration, sound amplitude, and spectral entropy. In all cases, we did not see a robust, acute change in song performance caused by the cutaneous stimulation. To ensure that the cutaneous stimulation on the scalp did not drive learning through an unexpected influence on brain activity in dorsal auditory areas of the pallium, we implanted the wire electrodes in the neck instead of the scalp in 7 out of 12 birds used in these experiments. The magnitude of vocal learning did not differ between the two groups of birds on any day of training ($0.679 < P_{boot} < 0.891$). To demonstrate that the ability to learn to adaptively shift the pitch of the target syllable is consistent across individual birds, we have plotted the results of 6 out of the 12 birds used in this dataset in *Figure 2—figure supplement 5* and *Figure 2—figure supplement 6* (*Figure 2—source data 2*). Also, we reanalyzed this dataset by measuring syllable pitch from syllables produced in the evening (6 PM to 8 PM) instead of in the morning, and we found no significant difference in the magnitude of learning between song collected between 10 AM and noon and song collected between 6 PM and 8 PM on any day of training (*Figure 2—figure supplement 7*, $0.167 < P_{boot} < 0.951$ on all days of training; *Figure 2—source data 3*). Taken together, these results indicate that the gradual, adaptive pitch shift is driven by non-auditory cutaneous stimulation and not by other unintentional effects of the stimulation, the methodology of wire implantation, or the time window of song analysis.

## LMAN is required for non-auditory vocal learning

We next investigated the neural circuitry that processes non-auditory feedback to drive vocal learning. To assess whether the AFP is required for non-auditory vocal learning, we measured the effect of lesions of LMAN, the output nucleus of the AFP, on learning magnitude in response to non-auditory

feedback (*Figure 1b*, Experiment 2). We performed cutaneous stimulation training experiments in the same individual birds before and after bilateral, electrolytic LMAN lesions (n = 5 birds, and one additional bird that only underwent postlesion training) or sham operations (*Figure 3a*, n = 5 birds). To perform cutaneous stimulation training in this group of experiments, we used the same protocol described previously, except we extended the period of cutaneous stimulation training an additional 2 days. During this extended training period, we set a new pitch threshold each morning to drive even greater amounts of learning ('staircase' training, see 'Materials and methods'). We did so in case LMAN lesions differentially impacted small and large magnitudes of learning, In adult songbirds with intact song systems (prelesion), such staircase training drove significant amounts of learning (*Figure 3c*).

We then lesioned LMAN and performed postlesion white noise training across conditions (LMAN lesion and sham) (*Figure 2—figure supplement 3b*). The efficacy of LMAN lesions was confirmed both by the presence of a characteristic reduction in the trial-to-trial variability of syllable pitch, analyzed across all labeled song syllables produced by the birds (including the target syllable used in learning experiments) (*Figure 3b*, *Figure 3—figure supplement 1a*, LMAN lesions p=0.002, sham lesions p=0.911, paired *t*-tests; *Figure 3—source data 1*; *Kao et al., 2005*; *Kao and Brainard, 2006*; *Hampton et al., 2009*), as well as by post-hoc histological measurements (see 'Materials and methods' and *Figure 3—figure supplement 2*). Following LMAN lesions, songbirds did not significantly change the pitch of the target syllable from baseline (zero) (the probability of resampled mean pitch on the final 4 days of training lesser than or equal to zero was $P_{boot} > 0.223$, n = 3). In contrast, following sham lesions, birds significantly changed the pitch of the target syllable in the adaptive direction (the probability of resampled mean pitch on the final 4 days of training days lesser than or equal to zero was $P_{boot} < 0.0010$, n = 5). This indicates that LMAN lesions induced significant deficits in auditory-driven vocal learning, consistent with previous work (*Hisey et al., 2018*).

LMAN lesions also significantly impaired non-auditory vocal learning. Prelesion, songbirds adaptively changed the pitch of the target syllable away from baseline in response to non-auditory feedback (the probability of resampled mean pitch on each day of cutaneous stimulation training lesser than or equal to zero was $P_{boot} < 0.0010$) (*Figure 3d*, *Figure 3—source data 2*). Postlesion, non-auditory vocal learning was abolished in those same birds (the probability of resampled mean pitch on each of the final 4 days of training lesser than or equal to zero was $0.297 < P_{boot} < 0.660$, where $0.025 < P_{boot} < 0.975$ indicates no significant difference, n = 6 birds; one bird from this group did not undergo prelesion experimentation) (*Figure 3d*, *Figure 3—source data 2*). Learning magnitude prelesion was significantly greater compared to learning magnitude postlesion ($P_{boot} < 0.007$ on each of the final 4 days of training). We observed significant learning during cutaneous stimulation training in both pre- and postsham-lesioned datasets (*Figure 3e*, for both presham and postsham datasets, the probability of resampled mean pitch on each day of cutaneous stimulation training lesser than or equal to zero was $P_{boot} < 0.0010$, n = 5 birds; *Figure 3—source data 3*). Also, the learning magnitudes during cutaneous stimulation training did not significantly differ in pre- vs. postsham datasets (the probability of resampled mean pitch of presham data on each day of training lesser than or equal to resampled mean pitch of postlesion data was $0.120 < P_{boot} < 0.524$). The amount of pitch change during cutaneous stimulation training for each individual experiment is shown in *Figure 2—figure supplement 2c, d*.

We also directly compared the lesion-induced change in learning magnitudes between conditions (LMAN lesion vs. sham) (*Figure 3—figure supplement 1b and c*). First, we calculated learning magnitude at the end of the fixed threshold training period across conditions. The lesion-induced change in learning magnitude (post – pre) for LMAN-lesioned birds was significantly greater than that for sham-operated birds (*Figure 3—figure supplement 1b*; two-sample Kolmogorov–Smirnov test, p=0.036, n = 5 birds in both groups). Next, we calculated learning magnitude at the end of the extended 'staircase' portion of cutaneous stimulation training across conditions. The lesion-induced change in learning magnitude (post – pre) for LMAN-lesioned birds calculated at this time point was also significantly greater than for sham-lesioned birds (*Figure 3—figure supplement 1c*; two-sample Kolmogorov–Smirnov test, p=0.004, n = 5 birds in both groups). These results show that LMAN is required for non-auditory vocal learning in adult songbirds, indicating that both auditory and non-auditory sensory feedback engage the AFP to drive adaptive changes to song.

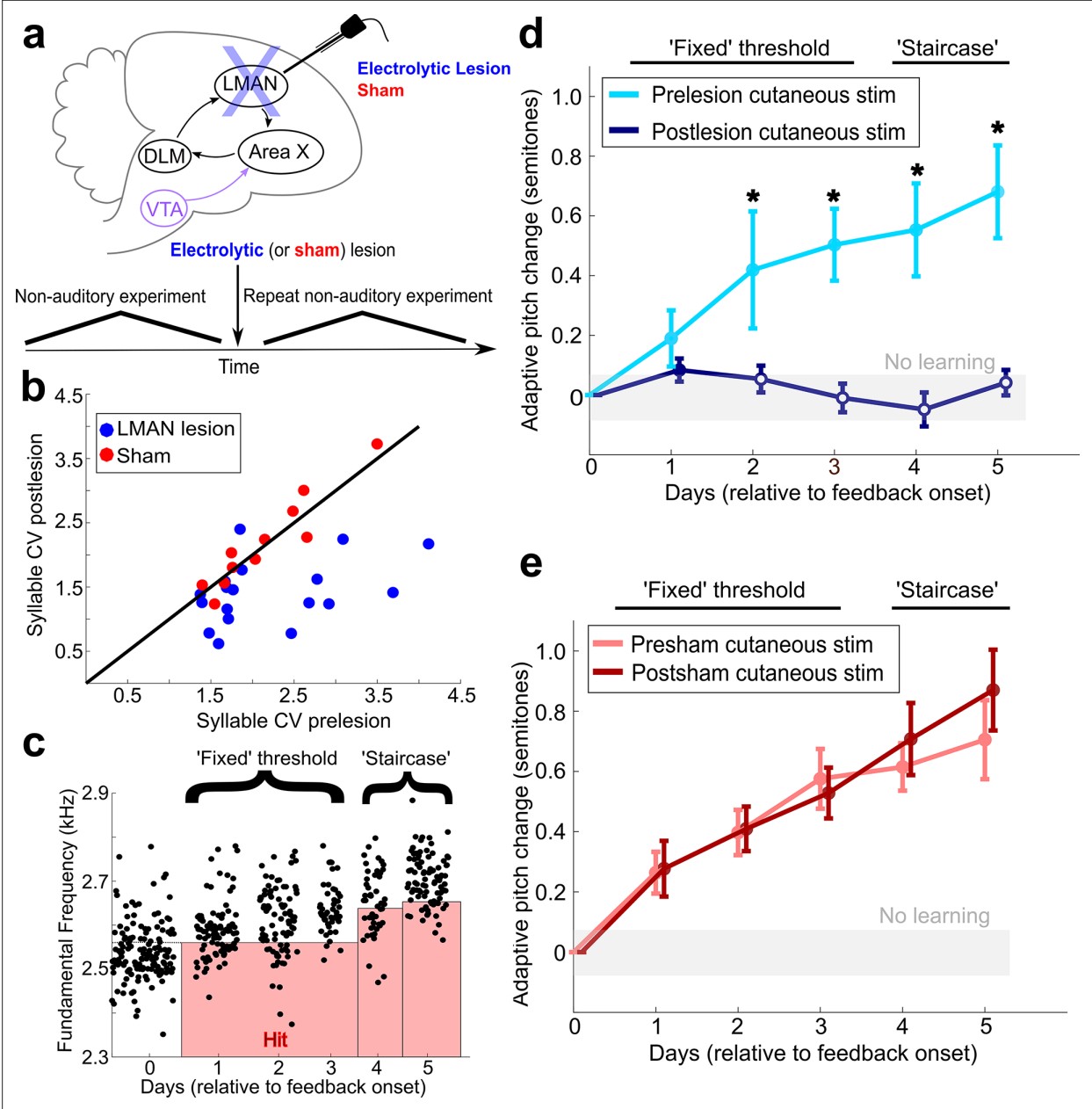

**Figure 3.** LMAN is required for non-auditory vocal learning. (**a**) Timeline for electrolytic lesions of LMAN and sham operations. (**b**) CV of syllable pitch pre- vs. postlesion and pre- vs. postsham. LMAN lesions induced a significant reduction in pitch CV, sham operations did not (paired *t*-tests, p=0.002, p=0.911, respectively). (**c**) Prelesion experiment. Training consisted of 3 days using a fixed pitch threshold, then additional days where the threshold was changed each morning ('staircase'). Each dot represents the pitch of a rendition of the target syllable. (**d**) Adaptive pitch change (in semitones) during cutaneous stimulation training (n = 6 LMAN-lesioned birds). Prelesion learning magnitude was significantly greater than baseline (the probability of resampled mean pitch on each day of training lesser than or equal to zero was $P_{boot} < 0.0010$, indicated by filled circles). Postlesion learning magnitude did not significantly differ from baseline ($0.297 < P_{boot} < 0.660$ on each of the final 4 days of training). Prelesion learning magnitude was significantly greater than postlesion learning magnitude (the probability of resampled mean pitch of prelesion data on the final 4 days of training lesser than or equal to resampled mean pitch of postlesion data was $P_{boot} < 0.0070$, indicated by asterisks). (**e**) Adaptive pitch change during cutaneous stimulation training (n = 5 sham-operated birds). Learning magnitudes were significantly greater than baseline both pre- and postsham (the probability of resampled mean pitch on each day of training lesser than or equal to zero was $P_{boot} < 0.0010$, indicated by filled circles). Learning magnitudes pre- vs. postsham did not significantly differ ($0.120 < P_{boot} < 0.524$ on all days of training).

The online version of this article includes the following source data, source code, and figure supplement(s) for figure 3:

**Source data 1.** Source data for analysis in *Figure 3B*.

**Source data 2.** Source data for analyses in *Figure 3C, D*.

*Figure 3 continued on next page*

*Figure 3 continued*

**Source data 3.** Source data for analysis in *Figure 3E*.

**Source code 1.** Source code for use with *Figure 3—source data 1* for analysis in *Figure 3B*.

**Source code 2.** Source code for use with *Figure 3—source data 2* for analysis in *Figure 3C*.

**Source code 3.** Source code for use with *Figure 3—source data 2* for analysis in *Figure 3D*.

**Source code 4.** Source code for use with *Figure 3—source data 3* for analysis in *Figure 3E*.

**Figure supplement 1.** Direct comparison of CV changes and learning changes between sham and lesioned (LMAN lesioned and 6-OHDA lesioned) groups.

**Figure supplement 2.** LMAN lesion histological analysis.

## Dopaminergic input to Area X is required for non-auditory vocal learning

We next assessed dopaminergic contributions to non-auditory vocal learning (*Figure 1b*, Experiment 3). Learning magnitude during cutaneous stimulation training was assessed before and after bilaterally lesioning dopaminergic projections in Area X, the basal ganglia nucleus of the AFP, in individual songbirds (*Figure 4a*, n = 5 birds). Selective lesions of dopaminergic projections in Area X were performed via bilateral 6-OHDA injections in Area X (see 'Materials and methods'), and the effectiveness of the 6-OHDA injections at lesioning dopaminergic innervation in Area X was quantified (*Figure 4—figure supplement 1*). This approach has previously been shown to selectively lesion dopaminergic inputs to Area X without damaging non-dopaminergic cells (*Hoffmann et al., 2016*; *Saravanan et al., 2019*).

We again measured the variability of syllable pitch pre- and postlesion by calculating syllable CV. Dopaminergic lesions in Area X did not induce a significant change in syllable CV (*Figure 4b*; paired *t*-test, p=0.397; *Figure 4—source data 1*). Sham operations also did not induce a significant change in syllable CV (*Figure 4b*; paired *t*-test, p=0.531). The lesion-induced changes in syllable CV (post – pre) were not significantly different for 6-OHDA-lesioned birds than for sham-lesioned birds (*Figure 3—figure supplement 1d*; two-sample Kolmogorov–Smirnov test, p=0.054). This finding is consistent with prior work using similar 6-OHDA injections to lesion dopaminergic input to Area X *Hoffmann et al., 2016*. Prior work has suggested a link between dopamine in songbird AFP and the generation of variability in syllable pitch in adult songbirds (*Murugan et al., 2013*; *Sasaki et al., 2006*; *Leblois et al., 2010*). It is likely that the dopamine lesion methodology we used, which spares about 50% of the dopaminergic input to Area X *Hoffmann et al., 2016*, is insufficient to impair dopamine-mediated generation of syllable variability. The result that these dopamine lesions do not alter vocal variability establishes that any learning deficits observed following lesions of AFP circuits are not simply due to decreased pitch variability.

Depletion of dopaminergic input to Area X significantly impaired adaptive, non-auditory vocal learning. Prelesion, songbirds adaptively changed the pitch of the target syllable during cutaneous stimulation training (the probability of resampled mean pitch on each of the final 4 days of cutaneous stimulation training lesser than or equal to zero was $P_{boot} < 0.010$) (*Figure 4c*, *Figure 4—source data 2*). Postlesion, these same birds were not able to adaptively change the pitch of the target syllable during cutaneous stimulation training (the probability of resampled mean pitch on each of the first 4 days of training lesser than or equal to zero was $0.067 < P_{boot} < 0.553$; the probability of resampled mean pitch on the final day of training greater than or equal to zero was $P_{boot} < 0.0010$, n = 5 birds). Learning magnitude prelesion was significantly greater compared to learning magnitude postlesion (the probability of resampled mean pitch from prelesion dataset on each of the final 3 days of cutaneous stimulation training lesser than or equal to resampled mean pitch from postlesion dataset was $P_{boot} < 0.0010$). Both pre- and postsham, songbirds displayed significant amounts of learning during cutaneous stimulation training (*Figure 4d*; the probability of resampled mean pitch from the presham dataset on each day other than day 2 of cutaneous stimulation training lesser than or equal to zero was $P_{boot} < 0.0010$; the probability of resampled mean pitch from the postsham dataset on each day of cutaneous stimulation training lesser than or equal to zero was $P_{boot} < 0.0010$, n = 3 birds; *Figure 4—source data 3*). Also, the learning magnitudes during cutaneous stimulation training did not significantly differ pre- vs. postsham (the probability of resampled mean pitch of presham data on each day of training lesser than or equal to resampled mean pitch of postlesion data was $0.653 < P_{boot} < 0.931$).

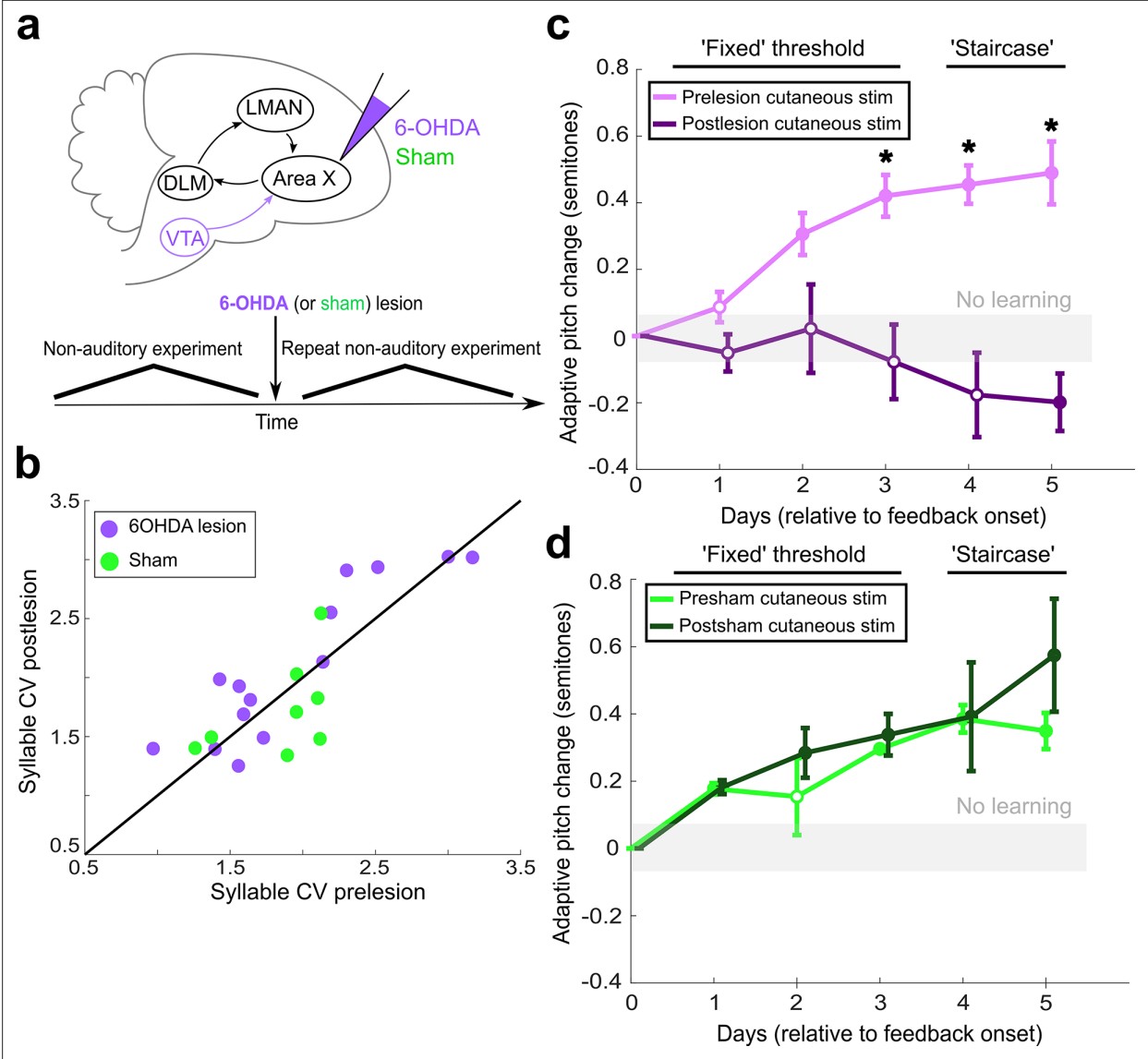

**Figure 4.** Dopaminergic input to Area X is required for non-auditory vocal learning. (**a**) Timeline for 6-OHDA and saline (sham) injections into Area X. (**b**) CV of syllable pitch pre- vs. postlesion and pre- vs. postsham. Neither dopamine lesions nor shams induced significant changes in pitch CV (paired *t*-tests, p=0.397 and p=0.531, respectively). (**c**) Adaptive pitch change (in semitones) during cutaneous stimulation training (n = 5 lesioned birds). Prelesion learning magnitude was significantly greater than baseline (the probability of resampled mean pitch on each of the final 4 days of training lesser than or equal to zero was $P_{boot} < 0.010$, indicated by filled circles). Postlesion learning magnitude did not significantly differ from baseline except for on the final day, when the mean changed in the anti-adaptive direction ($P_{boot} > 0.067$ on training days 1–4, $P_{boot} < 0.0010$ on training day 5). Prelesion learning magnitude was significantly greater than postlesion learning magnitude (the probability of resampled mean pitch from prelesion dataset on each of the final 3 days of training lesser than or equal to resampled mean pitch from postlesion dataset was $P_{boot} < 0.0010$, indicated by asterisks). (**d**) Adaptive pitch change (in semitones) during cutaneous stimulation training (n = 3 sham-lesioned birds). Learning magnitudes were significantly greater than baseline both pre- and postsham (the probability of resampled mean pitch from presham and postsham datasets on each day other than day 2 of training lesser than or equal to zero was $P_{boot} < 0.0010$, indicated by filled circles). Learning magnitudes pre- vs. postsham did not significantly differ ($0.653 < P_{boot} < 0.931$ on all days of training).

The online version of this article includes the following source data, source code, and figure supplement(s) for figure 4:

**Source data 1.** Source data for analysis in *Figure 4B*.

**Source data 2.** Source data for analysis in *Figure 4C*.

**Source data 3.** Source data for analysis in *Figure 4D*.

**Source code 1.** Source code for use with *Figure 4—source data 1* for analysis in *Figure 4B*.

**Source code 2.** Source code for use with *Figure 4—source data 2* for analysis in *Figure 4C*.

*Figure 4 continued on next page*

*Figure 4 continued*

**Source code 3.** Source code for use with *Figure 4—source data 3* for analysis in *Figure 4D*.

**Figure supplement 1.** 6-OHDA lesion histological analysis.

**Figure supplement 2.** Comparison of lesion magnitude and learning deficit.

These results demonstrate that dopaminergic input to Area X is required for adaptive changes in vocal output in response to non-auditory signals. The amount of pitch change during cutaneous stimulation training for each individual experiment is shown in *Figure 2—figure supplement 2d and e*.

## Discussion

Our results demonstrate that non-auditory feedback can drive vocal learning in adult songbirds, and that the AFP and its dopaminergic inputs are required for non-auditory vocal learning. We first demonstrated that adult songbirds learn to adaptively change the pitch of their song syllables in response to cutaneous stimulation (*Figure 1b*, Experiment 1). We next demonstrated that LMAN, the output nucleus of the AFP, is necessary for the expression of this non-auditory vocal learning (*Figure 1b*, Experiment 2). Finally, we showed that dopaminergic input to Area X, the basal ganglia nucleus of the AFP, is necessary for non-auditory vocal learning (*Figure 1b*, Experiment 3). These results show that adult vocal learning is not solely dependent on auditory feedback, and that the songbird AFP is not specialized just for processing auditory feedback for vocal learning, as has previously been hypothesized (*Murdoch et al., 2018*). Instead, these results, in conjunction with prior work using visual cues to drive changes in vocal output (*Zai et al., 2020*), indicate that the AFP processes auditory feedback as well as non-auditory feedback to drive vocal learning. Our results further show that the AFP processes somatosensory information to guide vocal learning, and that dopaminergic neural circuitry is necessary for non-auditory learning. Prior work has shown that songbird vocal muscles use somatosensory feedback to compensate for experimentally induced changes in respiratory pressure during song performance (*Suthers et al., 2002*). The result that the AFP underlies vocal learning driven by somatosensory signals (cutaneous stimulation) suggests that it could play a role in processing somatosensory information from vocal muscles to guide song performance. Also, the fact that mild cutaneous stimulation is different than the direct proprioceptive feedback from vocal muscles or vocal effectors, yet the AFP still underlies vocal learning in response to cutaneous stimulation, suggests that the AFP can integrate sensory information from a wide variety of sources of sensory feedback, even those not directly produced by vocalizations.

Our findings suggest the importance of neural pathways that convey non-auditory sensory signals to the song system. The neuroanatomical pathways for auditory feedback to enter the AFP are well-characterized. For example, recent work has demonstrated that songbird ventral pallidum (VP) receives input from auditory cortical areas, encodes auditory feedback information, and projects to VTA (*Chen et al., 2019*). This represents a likely pathway by which sensory information from white noise bursts could influence neural activity in VTA, which could then drive changes in the AFP that promote song learning. Comparatively less is known about the pathways in the songbird brain that might carry sensory information from cutaneous stimulation to the AFP. The results showing that dopaminergic input to Area X (which originates in the VTA) is necessary for non-auditory vocal learning suggest that pathways for non-auditory information ultimately project to the VTA, where this information could be encoded and transmitted to the AFP to drive learning. Further work is necessary to fully reveal the role the dopaminergic system plays in guiding non-auditory vocal learning. Following dopamine depletions in Area X, songbirds displayed a small but significant anti-adaptive change in syllable pitch. We believe this finding should be treated with caution because two of the four postlesion experiments in this dataset had to be stopped earlier than expected due to pandemic-related disruptions, and the extent to which the change in significance on day 5 might reflect this issue is unclear. We believe follow-up studies should aim to confirm this result in additional songbirds.

Prior studies have hinted that non-auditory feedback may play an important role in shaping vocalizations in ethological contexts, particularly during development. For example, juvenile songbirds that receive both auditory and visual feedback from live tutors display more accurate copying of tutor songs relative to juvenile songbirds that only receive auditory feedback from their tutors (*Chen et al., 2016*). Also, visual displays from adult song tutors positively reinforce the acquisition

of specific song elements in juvenile songbirds (*West and King, 1988*), further suggesting an important role for visual signals in social interactions during song learning. Our results that cutaneous stimulation can drive adaptive vocal changes in adult songbirds demonstrate that non-auditory signals, even in the absence of any social cues or other reinforcing sensory signals, can drive vocal learning with similar effectiveness as auditory feedback. Further, our work suggests that the AFP might play a role in processing non-auditory sensory information important to other social behaviors that involve vocal communication, such as courtship, territorial displays, and pair bonding.

It has been hypothesized that a key function of the songbird AFP circuitry is to encode auditory performance error: the evaluation of the match between the auditory feedback the songbirds receive and their internal goal for what their song should sound like (based on their stored memory of the tutor song template) (*Sober and Brainard, 2009*; *Saravanan et al., 2019*; *Fee and Goldberg, 2011*; *Gadagkar et al., 2016*). It has been difficult to determine the extent to which distorted auditory feedback drives adaptive changes in vocal output due to the aversive nature of the stimulus as opposed to the stimulus being interpreted by the bird as an auditory performance error. Some auditory vocal learning experiments have provided white noise bursts during ongoing song performance. In these experiments, songbirds adaptively modify their vocal output to avoid triggering white noise bursts as frequently (*Hoffmann et al., 2016*; *Tumer and Brainard, 2007*; *Charlesworth et al., 2011*). Also, white noise bursts can often cause song interruptions at first, suggesting that they are startling to the birds (*Hoffmann et al., 2016*; *Tumer and Brainard, 2007*). Other experiments have used distorted elements of song syllable segments played during song performance (distorted auditory feedback) and found that they elicit a pattern of activity in dopaminergic neurons consistent with the encoding of performance error (*Gadagkar et al., 2016*). Importantly, when bursts of noise are provided in non-vocal contexts, such as when a songbird stands on a particular perch (not during song performance), they can positively reinforce place preference (*Murdoch et al., 2018*). Thus, due to the various nuances in experimental methodology and the inherent difficulty in measuring the aversive nature of the auditory stimuli, it is unclear whether white noise bursts drive learning because the white noise is registered by the birds as a performance error or because the white noise is generally aversive. Although the results of the experiments described here do not directly address this, they do show that cutaneous stimulation (an explicit, external, aversive sensory stimulus) is sufficient to drive vocal learning. That the AFP underlies non-auditory learning suggests that the AFP does not solely encode auditory performance error. Instead, the AFP may encode more general information about whether vocal performance resulted in a 'good' or 'bad' outcome, and it may use this information to drive changes to future motor output.

The numerous analogies between the specialized vocal learning neural circuits that have evolved in songbirds and in humans suggest that our findings may be relevant to understanding the neural circuit mechanisms underlying human speech (*Doupe and Kuhl, 1999*; *Jarvis, 2019*; *Brainard and Doupe, 2002*; *Brainard and Doupe, 2013*). Human speech depends on both auditory and non-auditory sensory information to guide learning, yet very little is known about the neural mechanisms for non-auditory vocal learning (*Goldstein et al., 2003*; *Locke and Snow, 2010*; *Kuhl, 2007*). Our findings show that specialized vocal learning circuitry in songbirds processes non-auditory information to drive vocal plasticity. We suggest that the analogous vocal circuitry in humans may also underlie non-auditory vocal learning. This neural circuitry in humans may underlie the processing of multimodal sensory signals during social interactions that modulate speech learning (*Goldstein et al., 2003*; *Locke and Snow, 2010*; *Kuhl, 2007*), or the non-auditory, somatosensory feedback from vocal effectors during speech production (*Tremblay et al., 2003*).

## Materials and methods

All subjects were adult (>100 days old) male Bengalese finches (*Lonchura striata* var. *domestica*). All procedures were approved by Emory University's Institutional Animal Care and Use Committee (protocol #201700359). All singing was undirected (in the absence of a female bird) throughout all experiments.

## Delivery of non-auditory sensory feedback

To deliver non-auditory feedback signals to freely behaving songbirds during ongoing song performance, we first performed a surgery prior to any experimentation. Stainless steel wires were uninsulated at the tip (2–4 mm) and implanted subcutaneously on the bird's scalp. The approximate location of the scalp electrodes was 4.47 mm lateral and 6.3 mm anterior, relative to $Y_0$, far from the coordinates used for targeting auditory pallium, which are 1.1 mm anterior and 0.7 mm lateral, relative to $Y_0$, and 1.5 mm ventral from the surface of the brain (*Spool et al., 2021*). In 7 out of all 28 birds used across all experiments performed, wires were implanted intramuscularly in the birds' necks instead of on their scalps. The wires were soldered onto a custom-made circuit board that, during surgery, was placed on the bird's skull using dental cement. The circuit was connected to an electric stimulator (A-M Systems Isolated Pulse Stimulator), which produced pitch-contingent electrical currents through the wires implanted on the bird. We set the duration of cutaneous stimulation to 50 ms, which was a long enough duration to overlap with a large portion of the targeted syllable, yet a short enough duration to avoid interfering with following song syllables. We typically set the magnitude of electric current used for producing the cutaneous stimulation to 100–350 µA, which is behaviorally salient (the first few instances of cutaneous stimulation interrupt song), yet subtle enough as to not produce any body movements or signs of distress. Importantly we tried to match the same level of behavioral saliency across birds and, although the magnitude of electric current varied by a large amount across individual birds, it only varied by small amounts (<20 µA) pre- vs postlesion in those sets of experiments. Stimulations typically occurred within 20–30 ms of target syllable onset. Acute effects of electrical cutaneous stimulation on song structure, such as pitch, sound amplitude, entropy, or syllable sequence, were assessed to ensure these non-auditory stimuli produced no immediate, systematic, acoustic effects. This ensures that any observed gradual changes to song structure in response to cutaneous stimulation are due to non-auditory learning.

## Vocal learning paradigm and song analysis

Experimental testing of vocal learning was performed by driving adaptive changes in the fundamental frequency (pitch) of song syllables. To do so, we delivered pitch-contingent, non-auditory feedback (mild cutaneous electrical stimulation) to freely behaving songbirds in real time during song performance. We followed the same experimental protocols as experiments using white noise feedback to drive vocal learning (*Hoffmann et al., 2016*; *Tumer and Brainard, 2007*), except we used cutaneous stimulation instead of white noise bursts. After surgically implanting the fine-wire electrodes, we recorded song continuously for 3 days without providing any experimental feedback (cutaneous stimulation or white noise bursts). We refer to this period as 'baseline' (*Figure 2a*).

On the last (third) day of baseline, we measured the pitch of every rendition of the target syllable sung between 10 AM and 12 PM. We set a fixed pitch threshold based on the distribution of these pitches, such that we would provide sensory feedback only when the pitch of a rendition of the target syllable was above the 20th percentile of the baseline distribution ('hit'), and all renditions outside of this range did not trigger any feedback ('escape'). In this case, an adaptive vocal change would therefore be to change the pitch of the target syllable down, thereby decreasing the frequency of triggering cutaneous stimulation. In other experiments, we triggered feedback on all renditions below the 80th percentile of the baseline pitch distribution. In this case, an adaptive vocal change would be to change the pitch of the target syllable up. For each experiment, we randomly selected which of these two contingencies we employed so we could assess bidirectional adaptations in vocal motor output. In a subset of experiments, we used the 90th percentile and 10th percentile pitch values to set the pitch threshold. Importantly, we also randomly withheld triggering feedback on 10% of syllable renditions, regardless of syllable pitch or the experimental pitch contingency. This allows us to compare syllable renditions that did or did not result in cutaneous stimulation to assess any acute effects of this form of feedback on syllable structure.

At 10 AM on the fourth day of continuous song recording, we began providing pitch-contingent cutaneous stimulation in real time, targeted to specific song syllables sung within a specified range of pitches. We refer to this time period as 'cutaneous stimulation training' (*Figure 2a*). We used custom LabVIEW software to continuously record song, monitor song for specific elements indicative of the performance of the target syllable, perform online, rapid pitch calculation, and trigger feedback in real time. The computers running this software were connected to an electric stimulator. When the

electric stimulator received input from the LabVIEW software, it would then trigger a 50 ms burst of electric current through the implanted wire electrodes. During cutaneous stimulation training, we continuously recorded song and provided pitch-contingent cutaneous stimulation at the set fixed pitch threshold for 3 days. During these 3 days, every time the bird sang within the 'hit' range, a mild cutaneous stimulation was immediately triggered.

After 3 days of cutaneous stimulation training, we stopped providing cutaneous stimulation but continued recording unperturbed song for six additional days. We refer to this period as 'washout' (*Figure 2a*). During washout, we consistently observed spontaneous pitch restoration back to baseline across all experiments, which is in congruence with results from numerous white noise learning experiments (*Hoffmann et al., 2016*; *Andalman and Fee, 2009*; *Tumer and Brainard, 2007*). This allows for multiple experiments to be performed from similar baseline conditions in the same individual songbird.

In 14 out of all 28 birds used throughout this study, we performed both white noise training and cutaneous stimulation training in the same individual birds (*Figure 2a*). After the end of cutaneous stimulation training and 6 days of washout (when the pitch of the target syllable had restored to baseline levels), we performed the exact same experimental protocol, but we used white noise feedback instead of cutaneous stimulation. We could then compare learning in response to two different sources of sensory feedback in the same individual subject. We also sometimes reversed the order of experimentation by performing white noise experiments first and cutaneous stimulation experiments second. The order of experimentation was randomly decided for each songbird before beginning any white noise or cutaneous stimulation training.

For all LMAN lesion (*Figure 3a*) and 6-OHDA lesion experiments (*Figure 4a*), we performed a cutaneous stimulation training experiment prelesion. After 6 days of washout, we then performed surgery to lesion the neural circuit of interest. We then performed another cutaneous stimulation experiment in the same individual bird using the exact same protocol we used prelesion. For all of these lesion cutaneous stimulation experiments, we used the aforementioned cutaneous stimulation training paradigm, but with one slight alteration: we extended the number of days of cutaneous stimulation training and introduced a new methodology for setting the pitch threshold on these extended days of training. We still set a fixed pitch threshold based on analysis of the pitch distribution from the final day of baseline and performed 3 days of cutaneous stimulation training using this fixed pitch threshold. We refer to this portion of the lesion experiments as 'fixed' because the pitch threshold for determining whether a cutaneous stimulation was provided remained the same for all 3 days. Rather than stopping cutaneous stimulation training at this point, we instead continued providing pitch-contingent cutaneous stimulation for an additional 1–5 days. In the morning (at 10 AM) on each of these extended days of cutaneous stimulation training, we changed the pitch threshold to the 20th or 80th percentile (consistent with the initial contingency) of the pitch distribution of all renditions of the target syllable sung between 8 AM to 9:30 AM on that same day. As the bird changed the pitch of the target syllable in the adaptive direction, the new pitch thresholds continued to be set further and further in the adaptive direction to drive greater amounts of learning. We refer to these additional days as 'staircase.' After 1–5 days of staircase training, we stopped providing cutaneous stimulation and began the washout portion of the experiment. We used this experimental approach for both prelesion and postlesion experiments in our LMAN, 6-OHDA, and sham datasets. Importantly, although the number of days of staircase varied between individual birds, for each individual bird we matched the same number of prelesion days of staircase and postlesion days of staircase to ensure that, in both experimental conditions, the bird had an equivalent amount of time and opportunity to learn.

Custom-written MATLAB software (The MathWorks) was used for song analysis. On each day of every experiment, we quantified important song features, such as the pitch, sound amplitude, and spectral entropy, of all renditions of the targeted syllable produced between 10 AM and 12 PM. We did so to account for potential circadian effects on song production. We also reassessed our results (shown in *Figure 2*) by analyzing only syllable renditions produced between 6 PM and 8 PM using new methods for automated labeling of song syllables (*Cohen et al., 2022*). We found no statistically significant difference in learning magnitudes between the two forms of analysis (*Figure 2—figure supplement 7a*, $0.167 < P_{boot} < 0.951$ on all days of training). To ensure a level of consistency in the number of target syllable renditions measured on each day of an experiment, and to have a minimum number

of syllable renditions necessary to get an accurate measure of average syllable pitch, we checked that at least 30 renditions of the target syllable were sung within the 10 AM to 12 PM window. If there were less than 30 renditions of the target syllable, we extended the time window for song analysis by 1 hr in both directions (9 AM to 1 PM) and then reassessed to see if there were at least 30 syllable renditions. If not, we continued this process of extending the time window by 1 hr until 30 song renditions were in that day's dataset. Daily targeting sensitivity (hit rate) and precision (1 – false-positive rate) were measured in all experiments to ensure accurate targeting of the specific target syllable (and not accidentally targeting different song syllables). During the pitch-contingent feedback portion of the experiment, a subset (10%) of randomly selected target syllables did not trigger feedback, regardless of syllable pitch. These 'catch trials' allowed for the quantification and comparison of syllable features, such as pitch, sound amplitude, and entropy between trials when feedback was provided and trials when feedback was not provided. Pitch changes were quantified in units of semitones as follows:

$$s \;=\; 12 * \log_2 \; (h / b) \tag{1}$$

where s is the pitch change (in semitones) of the syllable, h is the average pitch (in Hertz) of the syllable, and b is the average baseline pitch (in Hertz) of the syllable.

## Analysis of variability in syllable pitch

We compared pitch variability pre- and postlesion using methods described in prior literature (*Kao et al., 2005*; *Kao and Brainard, 2006*; *Hampton et al., 2009*.) We analyzed all song renditions (within the 10 AM to 12 PM time window) performed on the final day of baseline prelesion and on the final day of baseline postlesion. We did so in our LMAN lesion experimental group as well as our 6-OHDA lesion experimental group. To measure the variability in pitch of the song syllables, we calculated the coefficient of variation (CV) for the pitch of each syllable using the following formula: CV = (Standard Deviation/Mean) * 100.

## LMAN lesions

Birds were anesthetized under ketamine and midazolam and were mounted in a stereotax. The beak angle was set to 20° relative to the surface level of the surgery table. For stereotactic targeting of specific brain regions (in this case, LMAN), anterior-posterior (AP) and medial-lateral (ML) coordinates were found relative to $Y_0$, a visible anatomical landmark located at the posterior boundary of the central venous sinus in songbirds. Dorsal-ventral (DV) coordinates were measured relative to the surface of the brain. Bilateral craniotomies were made at the approximate AP coordinates 4.9–5.7 mm and ML coordinates 1.5–2.5 mm. A lesioning electrode was then inserted 1.9–2.1 mm below the brain surface. These stereotactic coordinates targeted locations within LMAN. We then passed 100 µA of current for 60–90 s at 5–6 locations in LMAN in both hemispheres in order to electrolytically lesion the areas. This methodology was based on prior work involving LMAN lesions and LMAN inactivations (*Ali et al., 2013*; *Andalman and Fee, 2009*; *Kao et al., 2005*; *Kao and Brainard, 2006*; *Hampton et al., 2009*; *Warren et al., 2011*). In sham-operated birds, we instead performed small lesions in brain areas dorsal to LMAN. Again, this was consistent with methodology from prior studies (*Ali et al., 2013*; *Kao et al., 2005*; *Kao and Brainard, 2006*).

Birds recovered within 2 hr of surgery and began singing normally (at least 30 renditions of target syllable within 2 hr) typically 3–8 days after surgery. The number of songs sung per day did not differ significantly pre- vs. postlesion (paired *t*-test, p=0.249).

Behavioral measures indicated that LMAN was effectively lesioned in the birds in the LMAN lesion dataset. LMAN lesions in adult songbirds produce a significant decrease in the trial-to-trial variability of song syllable pitch (*Kao et al., 2005*; *Kao and Brainard, 2006*; *Hampton et al., 2009*). To assess lesion-induced changes in the variability of syllable pitch between conditions (LMAN lesion and sham), we calculated the CV of syllable pitch pre- and postlesion. We found that LMAN lesions induced a significant decrease in pitch CV (*Figure 3b*; paired *t*-test). Sham operations did not induce a significant change in syllable CV (*Figure 3b*; paired *t*-test, p=0.911). The lesion-induced changes in syllable CV (post – pre) were significantly greater than changes to CV in sham-lesioned controls (*Figure 3—figure supplement 1a*; two-sample Kolmogorov–Smirnov test, p=0.003).

Lesions were confirmed histologically using cresyl violet staining after completion of behavioral experimentation. In tissue from sham-operated birds, we identified Area X and LMAN based on

regions of denser staining as well as well-characterized anatomical landmarks (*Karten et al., 2013*). The histology methodology we employed followed previous literature involving LMAN lesions (*Ali et al., 2013*; *Kao et al., 2005*). We performed Nissl stains to stain for neuronal cell bodies in brain slices after experiments were complete (*Figure 3—figure supplement 2a*). We then calculated the optical density ratio of the region containing LMAN compared to background (a pallial region outside of LMAN) (*Figure 3—figure supplement 2b*; *Hoffmann et al., 2016*; *Saravanan et al., 2019*). The distribution of OD ratios from LMAN-lesioned tissue was significantly less than the OD ratios from sham-lesioned tissue (*Figure 3—figure supplement 2c*; two-sample Kolmogorov–Smirnov test, p<0.0010). This suggests that the density of neuronal cell bodies within LMAN was reduced following electrolytic lesions compared to following sham. Similar to a prior study, we also qualitatively assessed each slice of brain tissue to measure the percentage of intact LMAN remaining in the tissue (*Ali et al., 2013*). We found that all of the LMAN-lesioned birds had 80–100% of LMAN lesioned in both hemispheres.

## 6-OHDA lesions

Birds were anesthetized using ketamine and midazolam and were mounted in a stereotax, where the beak angle was set to 20° relative to the surface level of the surgery table. Isoflurane was used in later hours of the surgery to maintain an anesthetized state. Bilateral craniotomies were made above Area X from the approximate AP coordinates 4.5–6.5 mm and ML coordinates 0.75–2.3 mm relative to $Y_0$.

In each hemisphere, we inserted a glass pipette containing a 6-OHDA solution and made 12 pressure injections in a 3 mm × 4 mm grid between AP coordinates 5.1 mm and 6.3 mm, ML coordinates 0.9 mm and 2.2 mm, and the DV coordinate 3.18 mm relative to $Y_0$. Additional bilateral 6-OHDA injections were made at the AP coordinate 4.8 mm, ML coordinate ±0.8 mm, and DV coordinate 2.6 mm from the brain surface to lesion the most medial portion of Area X. Each injection consisted of 13.8 nL of 6-OHDA solution, injected at a rate of 23 nL/s at each site. The pipette was kept in place for 30 s after each injection and was then slowly removed. 6-OHDA solution was prepared using 11.76 mg 6-OHDA-HBr and 2 mg ascorbic acid in 1 mL of 0.9% normal saline solution. The solution was light-protected after preparation to prevent oxidation. In sham-operated birds, we performed the same surgical operations, except saline was injected into Area X instead of 6-OHDA. Again, this was consistent with methodology from prior studies (*Hoffmann et al., 2016*; *Saravanan et al., 2019*).

Birds recovered within 2 hr of surgery and began singing normally (at least 30 renditions of target syllable within 2 hr) typically 3–8 days after surgery. The number of songs sung per day did not differ significantly pre- vs. postlesion (paired *t*-test, p=0.290).

In order to confirm the effectiveness of 6-OHDA injections at lesioning dopaminergic input to Area X, we quantified the extent of the reduction of catecholaminergic fiber innervation within Area X after completing the behavioral experimentation in each bird (*Hoffmann et al., 2016*; *Saravanan et al., 2019*). To visualize dopaminergic innervation, we labeled tissue with a common biomarker for catecholaminergic cells (*Figure 4—figure supplement 1a*). To determine whether the concentration of dopaminergic fibers in Area X had decreased, we measured the optical density ratio (OD): the ratio of the stain density of Area X to the stain density of the surrounding striatum. OD ratios from individual 6-OHDA-lesioned brains decreased compared to control (*Figure 4—figure supplement 1b*). The distribution of all OD ratios from all of the 6-OHDA-lesioned tissue was significantly lower than that of the brain tissue from sham-operated birds (*Figure 4—figure supplement 1c*; two-sample Kolmogorov–Smirnov test, p<0.001). These results are similar to previous reports that used 6-OHDA injections to lesion dopaminergic input to Area X *Hoffmann et al., 2016*; *Saravanan et al., 2019*, and they indicate that the 6-OHDA injections successfully lesioned dopaminergic input to Area X.

Lesion size was quantified by determining the proportion of 6-OHDA-lesioned tissue that had an OD ratio of Area X to non-X striatum that was less than the fifth percentile of OD ratios in sham tissue. There was not a significant correlation between lesion size and the lesion-induced change in learning magnitude (post – pre) (*Figure 4—figure supplement 2a and b*; $R^2 = 0.019$, p=0.137).

## Histology

Between 14 and 54 days after surgery, birds were injected with a lethal dose of ketamine and midazolam and were perfused. The tissue was post-fixed in 4% paraformaldehyde at room temperature for 4–16 hr and then moved to a solution of 30% sucrose for at least 1 day at 4°C for cryoprotection.

Then, brain tissue was sliced in 40 µm sections. A chromogenic tyrosine hydroxylase (TH) stain was used to quantify the depletion of catecholaminergic fiber innervations in tissue collected from 6-OHDA-lesioned birds, and Nissl and fluorescent NeuN staining was used to assess the density of cell bodies in tissue from LMAN-lesioned and sham-operated birds. For one bird in the 6-OHDA-lesioned group, a Nissl stain was performed on alternate tissue sections to ensure no cell death occurred as a result of the lesion.

For TH immunohistochemistry, the tissue was incubated overnight in a primary anti-TH antibody solution. The tissue was next incubated in biotinylated horse anti-mouse secondary antibody solution for 1 hr. Then, the tissue was submerged in a diaminobenzidine (DAB) solution (two DAB tablets, Amresco E733 containing 5 mg DAB per tablet, 20 mL Barnstead $H_2O$, 3 µL $H_2O_2$) for less than 5 min for visualization. The DAB solution was prepared 1 hr prior to use. The tissue was washed, mounted, and coverslipped using Permount mounting medium.

### Tyrosine hydroxylase stain

Between each incubation, the tissue was washed with 0.1 M phosphate buffer (PBS) (23 g dibasic sodium phosphate, 5.25 g monobasic sodium phosphate, and 1 L deionized $H_2O$) three times for 10 min each. The tissue was first washed and then incubated in 0.3% $H_2O_2$ for 30 min and then 1% $NaBH_4$ for 20 min, followed by overnight incubation in a primary anti-TH antibody solution. The tissue was next incubated in biotinylated horse anti-mouse secondary antibody solution for 1 hr, then incubated in avidin-biotin-complex (ABC) solution for 1 hr that had been prepared 30 min prior to use. The tissue was then submerged in a DAB solution for less than 5 min. The tissue was then washed, mounted, and coverslipped using Permount mounting medium. These TH stains mark neurons expressing TH, which are catecholaminergic.

### Nissl stain

Tissue was washed in 0.1 M PBS three times for 10 min and was then mounted. The slides were incubated in Citrisolv twice for 5 min each, then delipidized in the following ethanol concentrations for 2 min each: 100, 100, 95, 95, and 70%. The tissue was briefly (less than 15 s) rinsed in deionized water, then incubated in cresyl violet (665 µL glacial acetic acid, 1 g cresyl violet acetate, and 200 mL deionized water) for 30 min. The tissue was rinsed in deionized water, then briefly (less than 15 s) submerged in the following ethanol concentrations for 2 min each: 70, 95, 95, 100, and 100%. The tissue was then incubated in Citrisolv twice for 5 min. The tissue was coverslipped using Permount mounting medium. These Nissl stains mark neuronal cell bodies.

### NeuN antibody stain

Between each incubation, the tissue was washed with 0.1 M PBS three times for 10 min each. The tissue was incubated in primary antibody solution (4 mL EMD Millipore guinea pig anti-NeuN Alexa Fluor 488 antibody, 6 mL Triton X-100, 20 mL normal donkey serum [NDS], and 1.95 mL 0.1 M PBS) overnight. The tissue was then washed and incubated in a secondary antibody solution (10 mL Jackson Labs donkey anti-guinea pig [DAG], 6 mL Triton X-100, and 1.975 mL 0.1 PBS) overnight. The tissue was then washed, mounted, and coverslipped with FluroGel mounting medium. Slides were sealed with lacquer. Images were taken under a widefield microscope (BioTek Lionheart FX, Sony ICX285 CCD camera, Gen5 acquisition software, ×1.25 magnification, 16-bit grayscale).

### Lesion analysis

Analysis of lesions was based on previously published methodology (*Hoffmann et al., 2016*; *Saravanan et al., 2019*). Images of stained tissue sections were obtained using a slide scanner and were converted into 8-bit grayscale images in ImageJ. In birds that received sham 6-OHDA lesions, Area X stains darker than surrounding striatum in TH-DAB-stained tissue due to a higher density of catecholaminergic inputs in Area X *Hoffmann et al., 2016*; *Saravanan et al., 2019*. The baseline level of stain darkness can vary from bird to bird. Therefore, rather than directly comparing the stain density of lesioned and sham tissue, the ratio of the stain density of Area X to that of the surrounding striatum (OD ratio) was calculated to determine whether the concentration of catecholaminergic fibers was decreased. Prior work demonstrated that the vast majority of catecholaminergic input to Area X is dopaminergic (*Hoffmann et al., 2016*).

For each section of tissue containing Area X, a customized ImageJ macro was used to select regions of interest (ROIs) within Area X and within a portion of striatum outside Area X by manually outlining Area X and selecting a circular 0.5-mm-diameter region of striatum anterior to Area X. Pixel count and OD of each ROI were measured, and the density of TH-positive fibers was calculated using the ratio of the OD of Area X to the OD of non-X-striatum.

The cumulative distribution of OD ratios for sham-operated birds was used to construct a 95% confidence interval and determine the threshold for lesioned tissue. 6-OHDA-lesioned tissue in which the OD ratio fell below the 5th percentile of control tissue had a significantly reduced TH-positive fiber density.

## Statistical testing

All error bars presented in the article represent SEM. When assessing whether a significant amount of vocal learning occurred in one experiment, we used one-sample $t$-tests to compare the mean pitch on the final day of training vs zero. To assess whether a significant difference in amount of learning occurred within an individual bird pre- vs. postlesion, we used paired $t$-tests. To assess significance between distributions of target syllable pitches on various days of the experiment (baseline, cutaneous stimulation training, washout), we used a two-sample Kolmogorov–Smirnov test.

Each experimental group had at least five birds, and for each bird, the target syllable was typically repeated well over 30 times a day. Therefore, the structure of our data is hierarchical, so error accumulates at different levels (birds and syllable iterations). Simply grouping all the data together ignores the non-independence between samples and underestimates the error. To address this issue, we employed a hierarchical bootstrap method to measure SEM and calculate p-values (*Saravanan et al., 2020*). For each experimental day, we calculated normalized pitch values (in semitones) (normalized to the mean pitch on the final baseline day during that particular experiment). We then generated a population of 10,000 bootstrapped means according to the following sampling procedure: to generate each individual subsample, we resampled across each level of hierarchy in our data (first resampled among the birds, then for each selected bird, we resampled among syllable iterations). The standard deviation of this population of bootstrapped means provides an accurate estimate of the uncertainty of the original data (*Saravanan et al., 2020*; *Saravanan et al., 2019*). Thus, the SEM values (which are used for error bars) we report when employing the hierarchical bootstrap method are equal to this standard deviation.

To calculate p-values and determine significance for comparing our data to zero using the hierarchical bootstrap method, we calculated $P_{boot}$: the proportion of bootstrapped means greater than zero compared to the total number of bootstrapped means. Using an acceptable type 1 error rate of 0.05, any value of this $P_{boot}$ ratio greater than 0.975 indicates the mean was significantly greater than zero and any value less than 0.025 indicates the mean was significantly less than zero. $P_{boot}$ values between 0.025 and 0.975 indicate no significant difference between the dataset and zero. Because we measure adaptive pitch changes in semitones, which are a normalized measure of pitch change where baseline is set to zero, this method of calculating $P_{boot}$ was employed in all instances where it was necessary to assess whether there was a significant change in pitch at the end of training compared to baseline (zero).

We also sometimes sought to determine significance for the comparison of two means rather than what was previously described (where we assess significance between one mean compared to baseline [zero]). We used a similar hierarchical bootstrap statistical methodology and calculated $P_{boot}$. The key difference is that, rather than measuring the proportion of resampled means greater than or less than zero, we instead calculate a joint probability distribution for the means of the two resampled datasets. We measured the percentage of this joint probability distribution that was above one side of the unity line. This percentage is the $P_{boot}$ value we report in these instances. If the proportion of this joint probability distribution that falls above the unity line is greater than 0.975, it indicated a significantly greater mean of dataset 1 over dataset 2. If the percentage of the joint probability distribution that was above the unity line was less than 0.025, it indicated a significantly lower mean of dataset 1 compared to dataset 2. $P_{boot}$ values between 0.025 and 0.975 indicate no significant difference between the two datasets. This method was employed in all instances where it was necessary to assess whether the learning magnitudes (adaptive pitch changes by the end of training) were significantly different pre- vs. postlesion (or pre- vs. postsham)

or across experimental conditions (e.g., postsham vs. postlesion or post-LMAN lesion vs. post-6-OHDA lesion).

In both forms of $P_{boot}$ calculation, the lowest statistical limit for $P_{boot}$ is $P_{boot} < 0.0010$, due to resampling $10^4$ times to create bootstrapped means. The highest possible limit for $P_{boot}$ is $P_{boot} > 0.9999$, for the same reason.

Sample sizes were not predetermined using a power analysis. Sample sizes of all sets of experiments were comparable to relevant prior literature (*Hoffmann et al., 2016*; *Tumer and Brainard, 2007*; *Saravanan et al., 2019*). If at any point during cutaneous stimulation training or white noise training a bird's rate of singing dropped below 10 songs per day for over 1 day, that experiment was stopped and the data were excluded from further analysis.

## Acknowledgements

This work was supported in part by NIH grants R01-EB022872, R01-NS084844, and R01-NS099375, a grant from the Simons Foundation as part of the Simons-Emory International Consortium on Motor Control, and HHMI.

## Additional information

### Funding

| Funder | Grant reference number | Author |
|---|---|---|
| National Institutes of Health | R01-EB022872 | James N McGregor |
| National Institutes of Health | R01-NS084844 | James N McGregor |
| National Institutes of Health | R01-NS099375 | James N McGregor |
| Simons Foundation | Emory International Consortium on Motor Control | Samuel J Sober |
| Howard Hughes Medical Institute | | Paul I Jaffe Michael S Brainard |

The funders had no role in study design, data collection and interpretation, or the decision to submit the work for publication.

### Author contributions

James N McGregor, Conceptualization, Data curation, Software, Formal analysis, Investigation, Visualization, Methodology, Writing - original draft, Writing – review and editing; Abigail L Grassler, Conceptualization, Data curation, Software, Formal analysis, Investigation, Methodology, Writing – review and editing; Paul I Jaffe, Conceptualization, Data curation, Validation, Methodology, Writing – review and editing; Amanda Louise Jacob, Formal analysis, Methodology; Michael S Brainard, Conceptualization, Resources, Supervision, Project administration, Writing – review and editing; Samuel J Sober, Conceptualization, Resources, Supervision, Funding acquisition, Writing - original draft, Project administration, Writing – review and editing

### Author ORCIDs

James N McGregor http://orcid.org/0000-0002-5187-0984
Paul I Jaffe http://orcid.org/0000-0003-0680-3923
Michael S Brainard http://orcid.org/0000-0002-9425-9907
Samuel J Sober http://orcid.org/0000-0002-1140-7469

### Ethics

All experimental protocols were approved by the Emory University and UC San Francisco Institutional Animal Care and Use Committees (protocol #201700359).

#### Decision letter and Author response

Decision letter https://doi.org/10.7554/eLife.75691.sa1

Author response https://doi.org/10.7554/eLife.75691.sa2

---

## Additional files

### Supplementary files

• Transparent reporting form

### Data availability

Source data are provided for all main figures and relevant figure supplements (Figure 2b–f, Figure 2—figure supplements 1–7, Figure 3b–e, Figure 3—figure supplement 1, and Figure 4b–d). MATLAB code for generating these figures is also provided in the associated source code files. Data and source code have also been uploaded to a public data repository on figshare, in a project titled 'Shared mechanisms of auditory and non-auditory vocal learning in the songbird brain'.

The following datasets were generated:

| Author(s) | Year | Dataset title | Dataset URL | Database and Identifier |
|---|---|---|---|---|
| McGregor J, Grassler A, Jaffe P, Jacob A, Brainard MS, Sober SJ | 2022 | McGregor_et_al_Figure_4_Source_data_3.mat. figshare. Dataset | https://doi.org/10.6084/m9.figshare.20183351.v1 | figshare, 10.6084/m9.figshare.20183351.v1 |
| McGregor J, Grassler A, Jaffe P, Jacob A, Brainard MS, Sober SJ | 2022 | McGregor_et_al_Figure_4_Source_data_1.mat. figshare. Dataset | https://doi.org/10.6084/m9.figshare.20183354.v1 | figshare, 10.6084/m9.figshare.20183354.v1 |
| McGregor J, Grassler A, Jaffe P, Jacob A, Brainard MS, Sober SJ | 2022 | McGregor_et_al_Figure_3_Source_Code_3.m. figshare. Software | https://doi.org/10.6084/m9.figshare.20183357.v1 | figshare, 10.6084/m9.figshare.20183357.v1 |
| McGregor J, Grassler A, Jaffe P, Jacob A, Brainard MS, Sober SJ | 2022 | McGregor_et_al_Figure_4_Source_Code_2.m. figshare. Software | https://doi.org/10.6084/m9.figshare.20183360.v1 | figshare, 10.6084/m9.figshare.20183360.v1 |
| McGregor J, Grassler A, Jaffe P, Jacob A, Brainard MS, Sober SJ | 2022 | McGregor_et_al_Figure_3_Source_data_1.mat. figshare. Dataset | https://doi.org/10.6084/m9.figshare.20183348.v1 | figshare, 10.6084/m9.figshare.20183348.v1 |
| McGregor J, Grassler A, Jaffe P, Jacob A, Brainard MS, Sober SJ | 2022 | McGregor_et_al_Figure_4_Source_Code_3.m. figshare. Software | https://doi.org/10.6084/m9.figshare.20183339.v1 | figshare, 10.6084/m9.figshare.20183339.v1 |
| McGregor J, Grassler A, Jaffe P, Jacob A, Brainard MS, Sober SJ | 2022 | McGregor_et_al_Figure_4_Source_data_2.mat. figshare. Dataset | https://doi.org/10.6084/m9.figshare.20183342.v1 | figshare, 10.6084/m9.figshare.20183342.v1 |
| McGregor J, Grassler A, Jaffe P, Jacob A, Brainard MS, Sober SJ | 2022 | McGregor_et_al_Figure_2_Source_data_3.mat. figshare. Dataset | https://doi.org/10.6084/m9.figshare.20183345.v1 | figshare, 10.6084/m9.figshare.20183345.v1 |
| McGregor J, Grassler A, Jaffe P, Jacob A, Brainard MS, Sober SJ | 2022 | McGregor_et_al_Figure_3_Source_Code_2.m. figshare. Software | https://doi.org/10.6084/m9.figshare.20183327.v1 | figshare, 10.6084/m9.figshare.20183327.v1 |

*Continued*

| Author(s) | Year | Dataset title | Dataset URL | Database and Identifier |
|---|---|---|---|---|
| McGregor J, Grassler A, Jaffe P, Jacob A, Brainard MS, Sober SJ | 2022 | McGregor_et_al_Figure_3_Source_data_2.mat. figshare. Dataset | https://doi.org/10.6084/m9.figshare.20183330.v1 | figshare, 10.6084/m9.figshare.20183330.v1 |
| McGregor J, Grassler A, Jaffe P, Jacob A, Brainard MS, Sober SJ | 2022 | McGregor_et_al_Figure_2_Source_data_2.mat. figshare. Dataset | https://doi.org/10.6084/m9.figshare.20183333.v1 | figshare, 10.6084/m9.figshare.20183333.v1 |
| McGregor J, Grassler A, Jaffe P, Jacob A, Brainard MS, Sober SJ | 2022 | McGregor_et_al_Figure_3_Source_Code_4.m. figshare. Software | https://doi.org/10.6084/m9.figshare.20183336.v1 | figshare, 10.6084/m9.figshare.20183336.v1 |
| McGregor J, Grassler A, Jaffe P, Jacob A, Brainard MS, Sober SJ | 2022 | McGregor_et_al_Figure_2_Source_data_1.mat. figshare. Dataset | https://doi.org/10.6084/m9.figshare.20183318.v1 | figshare, 10.6084/m9.figshare.20183318.v1 |
| McGregor J, Grassler A, Jaffe P, Jacob A, Brainard MS, Sober SJ | 2022 | McGregor_et_al_Figure_3_Source_data_3.mat. figshare. Dataset | https://doi.org/10.6084/m9.figshare.20183321.v1 | figshare, 10.6084/m9.figshare.20183321.v1 |
| McGregor J, Grassler A, Jaffe P, Jacob A, Brainard MS, Sober SJ | 2022 | McGregor_et_al_Figure_4_Source_Code_1.m. figshare. Software | https://doi.org/10.6084/m9.figshare.20183324.v1 | figshare, 10.6084/m9.figshare.20183324.v1 |
| McGregor J, Grassler A, Jaffe P, Jacob A, Brainard MS, Sober SJ | 2022 | McGregor_et_al_Figure_2_Source_code_3.m. figshare. Software | https://doi.org/10.6084/m9.figshare.20183315.v1 | figshare, 10.6084/m9.figshare.20183315.v1 |
| McGregor J, Grassler A, Jaffe P, Jacob A, Brainard MS, Sober SJ | 2022 | McGregor_et_al_Figure_2_Source_code_4.m. figshare. Software | https://doi.org/10.6084/m9.figshare.20183303.v1 | figshare, 10.6084/m9.figshare.20183303.v1 |
| McGregor J, Grassler A, Jaffe P, Jacob A, Brainard MS, Sober SJ | 2022 | McGregor_et_al_Figure_2_Source_Code_1.m. figshare. Software | https://doi.org/10.6084/m9.figshare.20183306.v1 | figshare, 10.6084/m9.figshare.20183306.v1 |
| McGregor J, Grassler A, Jaffe P, Jacob A, Brainard MS, Sober SJ | 2022 | McGregor_et_al_Figure_2_Source_Code_2.m. figshare. Software | https://doi.org/10.6084/m9.figshare.20183309.v1 | figshare, 10.6084/m9.figshare.20183309.v1 |
| McGregor J, Grassler A, Jaffe P, Jacob A, Brainard MS, Sober SJ | 2022 | McGregor_et_al_Figure_3_Source_Code_1.m. figshare. Software | https://doi.org/10.6084/m9.figshare.20183312.v1 | figshare, 10.6084/m9.figshare.20183312.v1 |

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
