## [Editor Report]

This is an important article that shows that songbirds can learn to adjust their song on the basis of somatosensory feedback, and not just auditory feedback as previously thought. Convincing evidence is provided that cutaneous stimulation-induced song learning requires the same dopamine-basal ganglia pathway previously implicated in natural auditory feedback-based learning, showing that vocal production circuits can flexibly learn from feedback from multiple modalities.

---

## [Decision Letter]

**Decision letter after peer review:**

Thank you for submitting your article "Shared mechanisms of auditory and non-auditory vocal learning in the songbird brain" for consideration by *eLife*. Your article has been reviewed by 3 peer reviewers, one of whom is a member of our Board of Reviewing Editors, and the evaluation has been overseen by Barbara Shinn-Cunningham as the Senior Editor. The following individual involved in review of your submission has agreed to reveal their identity: Nicolas Giret (Reviewer #3).

Essential revisions:

(1) Data analysis

1.1 Independently quantify distinct sites of electrical stimulation

Electrical stimulation of ~100 uAm for ~50ms could cause electric fields large enough to reach pallium underlying the skull, which could in turn discharge neurons in dorsal auditory areas of the avian pallium. Can the authors provide coordinates for their cutaneous wire implants with respect to known coordinates of auditory pallium? Also, the birds with neck cutaneous stimulation provide good controls for this concern. The authors should make absolutely clear in their figures which data came from neck stim and which came from scalp stim (e.g. in Figure 2, FigSupp1b,d,f; Figure 2,FigSupp2), and should include the justification for the neck stim in the main text.

1.2 Learning magnitude in cutaneous vs white noise

The authors also claim that there is no systematic difference between learning magnitudes of cutaneous stimulation and of auditory white noise stimulation, suggesting that both training methods result in the same learning efficacy. While their data indeed shows no significant difference between these training methods, there is little ground for this claim. First, learning magnitudes seem to vary a lot across individuals, they may be similar on average but there does not seem to be a correlation between the two. Second, similar learning magnitudes only show that the saliency of the two stimuli were adjusted to be roughly equal, which is not surprising given that they adjusted the magnitude of electric current using a similar criterion as in their initial 2007 paper: In (Tumer and Brainard 2007) they adjusted white noise amplitude until they observed stoppages during the first day of exposure, and in this manuscript they adjusted electric current to interrupt song on the first few instances of cutaneous stimulation.

In the distributions of the adaptive pitch changes between cutaneous and white noise feedback (Figure 2f) the sham and unoperated birds actually appear quite different: the unoperated seem to have more change their pitch when exposed to the white noise, while it is the opposite for the sham who seem to change more with the cutaneous stimulation. Could the authors provide some more statistics to justify the pooling of the two groups of birds?

1.3 Cases of sparse and/or noisy data lead to unconvincing claims

1.3.1. There are a few instances where more data would help to better evaluate the significance of the results. For example, only one of the three days of baseline song is shown and for only one example bird, and worst of all, the data is reduplicated in this bird on two days, which points to a serious flaw in either the analysis or the illustration. Authors should show more baseline days and include more birds.

1.3.2. The 2-sided KS test to assess the difference between baseline and end of cutaneous stimulation is extremely significant (10^-12) for that one example bird, which is nice, but it would be useful to see whether this is the case for all birds and not just that example bird on that example day. Also, it would be interesting to see how these statistics behave when comparing two or more baseline days. It is unlikely that the washout the KS analysis reveals in this one bird will apply in all birds.

1.3.3. Surprisingly, five days after the depletion of the DA inputs to the basal ganglia (Area X), there is a change of the pitch in the anti-adaptative direction that reaches statistical significance on day 5 (Figure 4c). This effect on the 5th day only might be related to the fact that the depletion of the DA spares about 50% of the inputs to Area X. But what could be the explanation for the change in the anti-adaptative direction?

1.4 Why exclude afternoon singing data?

Why was the analysis restricted to the song syllables that were produced between 10am and 12pm? What is the rationale for such a restriction? Did past papers on syllable contingent feedback driven pitch learning impose such a restriction? If not, why not? And why is it here? Are the results different when considering all the song syllables per day? This is a keypoint to show. Also, the reader only finds that information in the method section although it seems to me as an important one that needs to be provided in the main text. Finally, for the analysis only data between 10 am and 12 pm are used, this window is extended if birds sing less than 30 renditions of the target syllable during this time window. It is unclear from their description how often this is the case and how it influences their analysis. Furthermore, they exclude birds that dropped their singing rate below 10 songs per day for more than a day, again not stating how many birds were excluded based on this criterion.

(2) Failure to cite and consider Zai et al., 2020

It is an egregious oversight that the authors did not cite or discuss Zai et al., 2020. Both the ability of birds to learn from non-auditory stimuli and the involvement of the AFP in this process have been shown previously. This study showed that visual stimuli (short periods of light off) can successfully drive changes in pitch both in hearing and in deaf birds; furthermore, in deaf birds, the involvement of the AFP in this process has been shown using a similar lesioning approach. Thus, two out of the three main claims of novelty in the manuscript are not novel, despite the authors' claims. Thus, the main novelty beyond the 2020 study is that McGregor et al. are the first to show that somatosensory information (cutaneous electrical stimulation) can induce vocal plasticity and that dopaminergic projections to the AFP are somehow involved in this process.

(3) Interpretation of dopamine lesion experiments

The authors claim that dopaminergic input is necessary for observing adaptive changes, but their data suggests otherwise, namely that dopamine sets the direction of the change. Strictly speaking, the statement 'dopaminergic inputs are required for non-auditory vocal learning' is incorrect, since the data shows reversal in learning direction, which is a form of learning as well. Therefore, the apparent reversal in learning in DA-depleted conditions should be discussed.

(4) Effect of cutaneous stimulation on ongoing song

The absence of a transient effect of the electrical stimulation on the ongoing song (not only song stopping but also FM, pitch, entropy etc.) is claimed but not demonstrated. As the authors did quantify some important features (as stated in the methods, l. 567-568), some examples and analyses for at least one or two acoustic features should be shown (e.g. in a Supp Fig).

(5) Accurately contextualize distorted auditory feedback studies

The paragraph at line 420 is written as though all pitch contingent auditory feedback studies have been done with loud white noise bursts. But Andalman and Fee, 2009 and Chen et al., 2020 used broadband noise filtered in the 2-8kHz range so that it sounds like a zebra finch call (this noise actually elicits social calls and drives place preference, as cited (Murdoch et al., 2018)). And Gadagkar et al., 2016 additionally used displaced syllable fragments of each bird's own song at decibels less than what the singing bird would hear at the ear. These nuances of distinct feedback protocols are relevant to this paragraph and should be spelled out – as a loud non-social white noise burst with powers at low frequencies resembling knocks is likely perceived differently as a sound filtered to be in the bird's own vocalization range that is known to drive positive place preference and to evoke social calls. Please correct this paragraph so the papers are cited properly.

(6) Figure 2C: data from day 0 and day 1 are identical!

---

## [Author Response]

Essential revisions:(1) Data analysis1.1 Independently quantify distinct sites of electrical stimulationElectrical stimulation of ~100 uAm for ~50ms could cause electric fields large enough to reach pallium underlying the skull, which could in turn discharge neurons in dorsal auditory areas of the avian pallium. Can the authors provide coordinates for their cutaneous wire implants with respect to known coordinates of auditory pallium? Also, the birds with neck cutaneous stimulation provide good controls for this concern. The authors should make absolutely clear in their figures which data came from neck stim and which came from scalp stim (e.g. in Figure 2, FigSupp1b,d,f; Figure 2,FigSupp2), and should include the justification for the neck stim in the main text.

We thank the reviewers for this comment and have edited the manuscript to address their concerns. The approximate location of the scalp electrodes was 4.5 mm lateral and 6.3 mm anterior, relative to Y_0_. The coordinates used in prior literature for targeting auditory pallium are 1.1 mm anterior and 0.7 mm lateral, relative to Y_0_, and 1.5 mm ventral from the surface of the brain (Spool et al., 2021), which provides 6.1 mm of space between our electrodes and auditory pallium. We therefore believe the location of the scalp electrodes was far enough away from auditory pallium to avoid potentially discharging neurons in the region. We also agree that the neck stimulation also addresses this potential concern. We have included this information in the main manuscript text (pg. 6, lines 232-238, and pg. 15, lines 515-518):

“To ensure that the cutaneous stimulation on the scalp did not drive learning through an unexpected influence on brain activity in dorsal auditory areas of the pallium, we implanted the wire electrodes in the neck instead of the scalp in 7 out of 12 birds used in these experiments. The magnitude of vocal learning did not differ between the two groups of birds on any day of training (0.679<P_boot_<0.891). Taken together, these results indicate that the gradual, adaptive pitch shift is driven by non-auditory cutaneous stimulation and not by other unintentional effects of the stimulation.”

“The approximate location of the scalp electrodes was 4.47 mm lateral and 6.3 mm anterior, relative to Y_0_, far from the coordinates used for targeting auditory pallium, which are: 1.1 mm anterior and 0.7 mm lateral, relative to Y_0_, and 1.5 mm ventral from the surface of the brain”

Also, we have edited the relevant main text figure (Figure 2 f in the revised manuscript) to illustrate which results were from birds with neck-implanted electrodes and which were from birds with scalp-implanted electrodes:

Figure 2F shows that electric shocks delivered to the neck and scalp produced overlapping ranges of learning magnitude (x-axis values of filled and empty squares). Moreover, note that the largest learning magnitudes were observed in animals with electrodes located on the neck (i.e. farther from the auditory pallium from the scalp-implanted electrodes).

Finally, we have added a new panel to Figure 2, Figure Supplement 1, to demonstrate the results of cutaneous training from experiments using neck vs scalp electrodes (Figure 2, Figure Supplement 2 b in the revised manuscript). There was no statistical difference in magnitude of adaptive pitch change between the two groups on any day of cutaneous training (0.679<P_boot_<0.891).

1.2 Learning magnitude in cutaneous vs white noiseThe authors also claim that there is no systematic difference between learning magnitudes of cutaneous stimulation and of auditory white noise stimulation, suggesting that both training methods result in the same learning efficacy. While their data indeed shows no significant difference between these training methods, there is little ground for this claim. First, learning magnitudes seem to vary a lot across individuals, they may be similar on average but there does not seem to be a correlation between the two. Second, similar learning magnitudes only show that the saliency of the two stimuli were adjusted to be roughly equal, which is not surprising given that they adjusted the magnitude of electric current using a similar criterion as in their initial 2007 paper: In (Tumer and Brainard 2007) they adjusted white noise amplitude until they observed stoppages during the first day of exposure, and in this manuscript they adjusted electric current to interrupt song on the first few instances of cutaneous stimulation.

We thank the reviewer for their comments. We agree that learning magnitudes vary substantially across individuals, and that our data do not allow us to make strong conclusions about relative stimulus strength/ aversiveness across modalities within single birds or within a single modality across subjects. We have therefore revised our text to make it clear that, as stated by the Reviewer above, the results in Figure 2f serve to establish primarily that shock and white noise produce (by design) similar ranges of learning, and do not establish the extent to which sensitivity to different sensory modalities varies across individuals. We have edited the manuscript text to clarify this (pg. 5, lines 183-197):

“To further characterize cutaneous stimulation training and to compare this form of learning to well-established vocal learning paradigms, we performed multiple learning experiments – one cutaneous stimulation and one white noise – in 8 out of the 12 individual birds from this data set where the implanted electrode wires remained intact for a long enough time to perform multiple sets of experiments (Figure 2a). To account for the potential influence of multiple trainings in the same individual birds on magnitude of learning, we randomized the order of white noise training and cutaneous stimulation training for the birds who underwent both training paradigms. We also included 6 LMAN sham operated birds from a later set of experiments in this particular analysis. We did so because the sham operated birds had intact song systems and underwent both cutaneous stimulation and white noise training. Also, we found no statistically significant difference between the magnitude of learning by the end of training in birds who did not undergo craniotomies for LMAN, 6OHDA, or sham lesions compared with the magnitude of learning in birds that received sham LMAN lesions for either white noise experiments (2 sample t-test, p = 0.779) or cutaneous stimulation experiments (2 sample t-test, p = 0.148).”

In the distributions of the adaptive pitch changes between cutaneous and white noise feedback (Figure 2f) the sham and unoperated birds actually appear quite different: the unoperated seem to have more change their pitch when exposed to the white noise, while it is the opposite for the sham who seem to change more with the cutaneous stimulation. Could the authors provide some more statistics to justify the pooling of the two groups of birds?

As requested, we have added a statistical analysis of whether learning magnitude differences between sham and unoperated birds. We found no significant difference between the groups when comparing the magnitude of learning by the end of white noise training (2 sample KS-test, p = 0.779). We also analyzed the average magnitude of learning during cutaneous stimulation training from unoperated birds compared with the learning magnitude during stimulation training from birds that received sham LMAN lesions, and again found no statistically significant difference between the groups (2 sample KS-test, p = 0.148). We have added this information to the manuscript (pg. 5, lines 192-197):

“Also, we found no statistically significant difference between the magnitude of learning by the end of training in unoperated birds compared with the magnitude of learning in birds that received sham LMAN lesions for either white noise experiments (2 sample t-test, p = 0.779) or cutaneous stimulation experiments (2 sample t-test, p = 0.148).”

1.3 Cases of sparse and/or noisy data lead to unconvincing claims1.3.1. There are a few instances where more data would help to better evaluate the significance of the results. For example, only one of the three days of baseline song is shown and for only one example bird, and worst of all, the data is reduplicated in this bird on two days, which points to a serious flaw in either the analysis or the illustration. Authors should show more baseline days and include more birds.

We deeply apologize for this error and thank both Reviewers for bringing it to our attention. On investigation we discovered that this mistake was caused by an error in the figure plotting code only, did not reflect any errors in analysis, and did not affect any of the results reported. We have corrected the figure (Figure 2 C in the revised manuscript), and have added additional data from baseline days of song recording to demonstrate the stability of syllable pitch during baseline conditions.

To further address this concern, we have also added a new supplemental figure (Figure 2 —figure supplement 5 in the revised manuscript), where we show the pitch for all renditions of the target syllable between 10 – noon across every day of recording, including all days of baseline and cutaneous training, from 6 additional experiments, which we randomly selected from our dataset to illustrate the range of learning behavior across experiments. Also, we are providing full datasets for all experiments, along with the code to generate all figures.

1.3.2. The 2-sided KS test to assess the difference between baseline and end of cutaneous stimulation is extremely significant (10^-12) for that one example bird, which is nice, but it would be useful to see whether this is the case for all birds and not just that example bird on that example day. Also, it would be interesting to see how these statistics behave when comparing two or more baseline days. It is unlikely that the washout the KS analysis reveals in this one bird will apply in all birds.

To address this reviewer’s interest in seeing additional results to the example ones we chose for the main text figure, we have created a new Supplementary Figure (Figure 2 Supplement 6 in the revised manuscript), where we created the same CDF plot for multiple other example experiments from the dataset from the same birds shown above in Figure 2 Supplement 5 (see response to the above question). We report the result of KS-tests for each of these example experiments in the figure legend. We’ve also performed KS-tests comparing the data on the final day of baseline to the final day of cutaneous stimulation training for all of the experiments in this unoperated dataset. 11 out of the 12 experiments with 3 days of baseline recording resulted in a p-value < 0.05, and highly significant differences were the norm. The exact p-values for each experiment shown in Figure 2 Figure Supplement 6 are reported in the figure legend.

1.3.3. Surprisingly, five days after the depletion of the DA inputs to the basal ganglia (Area X), there is a change of the pitch in the anti-adaptative direction that reaches statistical significance on day 5 (Figure 4c). This effect on the 5th day only might be related to the fact that the depletion of the DA spares about 50% of the inputs to Area X. But what could be the explanation for the change in the anti-adaptative direction?

We agree with the reviewer that the observed anti-adaptive change in average pitch following dopamine lesions is interesting. However, although this difference achieves statistical significance, we believe that this finding should be treated with caution due to the fact that two of the four postlesion experiments had to be stopped early due to pandemic-related disruptions (Figure 2 Supplement 2, panel e in the revised manuscript, we have added this information to the figure legend.) The duration of the postlesion experiments differed due to the unexpected need to terminate experiments earlier than planned during the initial stages of the pandemic. It is therefore unclear the extent to which the change in significance on day 5 might reflect the removal of half of the subjects in the key condition. In preparing the manuscript, we considered excluding data from day 5 altogether due to these issues but decided to show the data to let readers make up their minds.

1.4 Why exclude afternoon singing data?Why was the analysis restricted to the song syllables that were produced between 10am and 12pm? What is the rationale for such a restriction? Did past papers on syllable contingent feedback driven pitch learning impose such a restriction? If not, why not? And why is it here? Are the results different when considering all the song syllables per day? This is a keypoint to show. Also, the reader only finds that information in the method section although it seems to me as an important one that needs to be provided in the main text. Finally, for the analysis only data between 10 am and 12 pm are used, this window is extended if birds sing less than 30 renditions of the target syllable during this time window. It is unclear from their description how often this is the case and how it influences their analysis. Furthermore, they exclude birds that dropped their singing rate below 10 songs per day for more than a day, again not stating how many birds were excluded based on this criterion.

We thank the reviewers for these important questions. Prior studies from our own and other groups have similarly restricted analyses to particular time intervals (Sober and Brainard, 2009, Ali et al., 2013). In this study, we initially restricted the analysis to song produced between 10 A.M. and 12 P.M. for the sake of convenience, since labeling all syllable renditions for every day from the large number of experiments performed throughout these studies would have been very time-consuming. Moreover, restricting analysis to the same period of the day helps to mitigate the potential impact of circadian cycles on song behavior. In Sober and Brainard, 2009, we directly addressed whether this time restriction impacted behavior in a vocal learning paradigm by comparing the data produced between 10 A.M. and 12 P.M. with the analysis of song syllables produced between 6 P.M. – 8 P.M. and showed no statistical difference in learning between the two. We have added this information to the main text of the manuscript (pg. 6, lines 239-243) in addition to the Methods section (pg. 16, lines 633-638). Further, we have added a new supplemental figure where we analyzed all syllables produced between 6-8 P.M., and compared these results to those obtained by analyzing songs produced between 10 A.M. – 12 P.M. (Figure 2—figure supplement 7 in the revised manuscript). We found no statistically significant difference in the magnitude of learning between the two groups, suggesting that restricting our analysis to this particular time of the day did not impact the main results described in the paper.

(2) Failure to cite and consider Zai et al., 2020It is an egregious oversight that the authors did not cite or discuss Zai et al., 2020. Both the ability of birds to learn from non-auditory stimuli and the involvement of the AFP in this process have been shown previously. This study showed that visual stimuli (short periods of light off) can successfully drive changes in pitch both in hearing and in deaf birds; furthermore, in deaf birds, the involvement of the AFP in this process has been shown using a similar lesioning approach. Thus, two out of the three main claims of novelty in the manuscript are not novel, despite the authors' claims. Thus, the main novelty beyond the 2020 study is that McGregor et al. are the first to show that somatosensory information (cutaneous electrical stimulation) can induce vocal plasticity and that dopaminergic projections to the AFP are somehow involved in this process.

We agree completely and apologize for this error. We have edited the manuscript text in multiple locations in the introduction and discussion to address this important prior study. Specifically, we have added additional information and citations on pg. 2, lines 81-83 and 88-94, and pg. 12 lines 418-422 (please see below for an example). We hope that the revised manuscript properly frames our study in the context of this important earlier study.

“Recent work has demonstrated that the songbird AFP receives anatomical projections from brain regions that process non-auditory sensory information^27^, and that Area X plays a crucial role in processing visual information to shape vocal output^17^, yet it remains unclear whether and how the AFP processes somatosensory feedback to drive vocal learning, and whether dopaminergic input to the AFP is involved in non-auditory forms of learning.”

(3) Interpretation of dopamine lesion experimentsThe authors claim that dopaminergic input is necessary for observing adaptive changes, but their data suggests otherwise, namely that dopamine sets the direction of the change. Strictly speaking, the statement 'dopaminergic inputs are required for non-auditory vocal learning' is incorrect, since the data shows reversal in learning direction, which is a form of learning as well. Therefore, the apparent reversal in learning in DA-depleted conditions should be discussed.

We thank the reviewer for this comment. Please see our response to 1.3.3 above, where we explain that we had to terminate a subset of experiments earlier than expected due to the pandemic and therefore do not wish to make any strong claims about the results on the final day of data collection for this particular experiment. We have edited the text of the revised manuscript to more carefully explain the differences in the expression of learning prelesion vs postlesion (pg. 11, lines 396-400):

“These results demonstrate that dopaminergic input to Area X is required for adaptive changes in vocal output in response to non-auditory signals. “

(4) Effect of cutaneous stimulation on ongoing songThe absence of a transient effect of the electrical stimulation on the ongoing song (not only song stopping but also FM, pitch, entropy etc.) is claimed but not demonstrated. As the authors did quantify some important features (as stated in the methods, l. 567-568), some examples and analyses for at least one or two acoustic features should be shown (e.g. in a Supp Fig).

We have included an additional supplemental figure to demonstrate the absence of a transient effect of cutaneous stimulation on ongoing song by assessing syllable pitch, entropy, volume, and duration (Figure 2—figure supplement 4 in the revised manuscript):

We discuss these results in the revised manuscript text (pg. 5, lines 216-232):

“To confirm that cutaneous stimulation learning was truly driven by the non-auditory stimulus and not by an unintentional, acute change in vocal output caused by the cutaneous stimulation, we measured the syllable features of interleaved “catch” trials, where cutaneous stimulation was randomly withheld (see Methods), on each day of cutaneous stimulation training. For each experiment, we normalized the pitch of each catch trial from each day of training to the mean pitch of all trials where cutaneous stimulation was provided. We excluded any experiments where the total number of catch trials was less than 10. In every case, the normalized catch trials did not differ significantly from 1, indicating that the pitch of catch trials were highly similar to trials where cutaneous stimulation was provided (Figure 2—figure supplement 4a; t-test, 0.071<p<0.997 for each experiment). For comparison, we also performed the same analysis on randomly selected trials from a day of baseline recording, where cutaneous stimulation was not provided on any trials (Figure 2—figure supplement 4a). There was no significant difference between this data set and the normalized catch trials (paired t-test, p = 0.339). We repeated this analysis for other syllable features, such as syllable duration, volume, and spectral entropy. In all cases, we did not see a robust, acute change in song performance caused by the cutaneous stimulation.”

(5) Accurately contextualize distorted auditory feedback studiesThe paragraph at line 420 is written as though all pitch contingent auditory feedback studies have been done with loud white noise bursts. But Andalman and Fee, 2009 and Chen et al., 2020 used broadband noise filtered in the 2-8kHz range so that it sounds like a zebra finch call (this noise actually elicits social calls and drives place preference, as cited (Murdoch et al., 2018)). And Gadagkar et al., 2016 additionally used displaced syllable fragments of each bird's own song at decibels less than what the singing bird would hear at the ear. These nuances of distinct feedback protocols are relevant to this paragraph and should be spelled out – as a loud non-social white noise burst with powers at low frequencies resembling knocks is likely perceived differently as a sound filtered to be in the bird's own vocalization range that is known to drive positive place preference and to evoke social calls. Please correct this paragraph so the papers are cited properly.

We thank the reviewer for this comment and agree that the nuances of these different auditory feedback experiments warrant further discussion. We have edited this paragraph in the text of the revised manuscript to address these important points (pg. 14, lines 474-498):

“It has been hypothesized that a key function of the songbird AFP circuitry is to encode auditory performance error: the evaluation of the match between the auditory feedback the songbirds receive and their internal goal for what their song should sound like (based on their stored memory of the tutor song template)^11,31,43,44^. It has been difficult to determine the extent to which distorted auditory feedback drives adaptive changes in vocal output due to the aversive nature of the stimulus as opposed to the stimulus being interpreted by the bird as an auditory performance error. Some auditory vocal learning experiments have provided white noise bursts during ongoing song performance. In these experiments, songbirds adaptively modify their vocal output to avoid triggering white noise bursts as frequently^24,29,45^. Also, white noise bursts can often cause song interruptions at first, suggesting they are startling to the birds^24,29^. Other experiments have used distorted elements of song syllable segments played during song performance (distorted auditory feedback), and found that they elicit a pattern of activity in dopaminergic neurons consistent with the encoding of performance error^44^. Importantly, when bursts of noise are provided in non-vocal contexts, such as when a songbird stands on a particular perch (not during song performance), they can positively reinforce place preference^38^. Thus, due to the various nuances in experimental methodology and the inherent difficulty in measuring the aversive nature of the auditory stimuli, it is unclear whether white noise bursts drive learning because the white noise is registered by the birds as a performance error or because the white noise is generally aversive. Although the results of the experiments described here do not directly address this, they do show that cutaneous stimulation (an explicit, external, aversive sensory stimulus) is sufficient to drive vocal learning. That the AFP underlies non-auditory learning suggests that the AFP does not solely encode auditory performance error. Instead, the AFP may encode more general information about whether vocal performance resulted in a “good" or “bad" outcome, and it may use this information to drive changes to future motor output.”

(6) Figure 2C: data from day 0 and day 1 are identical!

We thank the reviewer for this comment. Please see our response to 1.3.1, where we explain that this was caused by an error in the plotting and not an underlying issue with the data or analysis pipeline. We have edited the figure in the revised manuscript (Figure 2 c).